# Mechanical stress contributes to the expression of the *STM* homeobox gene in Arabidopsis shoot meristems

Benoît Landrein[1,2], Annamaria Kiss[1,2], Massimiliano Sassi[1], Aurélie Chauvet[1,2], Pradeep Das[1,2], Millan Cortizo[3,4], Patrick Laufs[3,4], Seiji Takeda[5], Mitsuhiro Aida[6], Jan Traas[1], Teva Vernoux[1], Arezki Boudaoud[1,2], Olivier Hamant[1,2]*

[1]Laboratoire de Reproduction de développement des plantes, INRA, CNRS, ENS Lyon, UCB Lyon 1, Université de Lyon, Lyon, France; [2]Laboratoire Joliot-Curie, Laboratoire de Physique, CNRS, ENS Lyon, UCB Lyon 1, Université de Lyon, Lyon, France; [3]INRA, Institut Jean-Pierre Bourgin, UMR 1318, ERL CNRS 3559, Saclay Plant Sciences, Versailles, France; [4]AgroParisTech, Institut Jean-Pierre Bourgin, UMR 1318, ERL CNRS 3559, Saclay Plant Sciences, Versailles, France; [5]Cell and Genome Biology, Graduate School of Life and Environmental Sciences, Kyoto Prefectural University, Kyoto, Japan; [6]Graduate School of Biological Sciences, Nara Institute of Science and Technology, Nara, Japan

**Abstract** The role of mechanical signals in cell identity determination remains poorly explored in tissues. Furthermore, because mechanical stress is widespread, mechanical signals are difficult to uncouple from biochemical-based transduction pathways. Here we focus on the homeobox gene *SHOOT MERISTEMLESS (STM)*, a master regulator and marker of meristematic identity in *Arabidopsis*. We found that *STM* expression is quantitatively correlated to curvature in the saddle-shaped boundary domain of the shoot apical meristem. As tissue folding reflects the presence of mechanical stress, we test and demonstrate that *STM* expression is induced after micromechanical perturbations. We also show that *STM* expression in the boundary domain is required for organ separation. While *STM* expression correlates with auxin depletion in this domain, auxin distribution and *STM* expression can also be uncoupled. *STM* expression and boundary identity are thus strengthened through a synergy between auxin depletion and an auxin-independent mechanotransduction pathway at the shoot apical meristem.

*For correspondence: Olivier. Hamant@ens-lyon.fr

**Competing interests:** The authors declare that no competing interests exist.

## Introduction

Almost 100 years ago, Thomas D'Arcy Wentworth Thompson proposed that shape of plants and animals could be described as a consequence of the laws of physics (*D'Arcy Thompson, 1917*). While this view has been overshadowed by the rise of molecular biology and morphogen-based patterning mechanisms, a growing contribution of mechanics in shape changes is currently emerging. In this framework, an instructing role of mechanical forces has been successfully explored theoretically (*Shraiman, 2005*; *Aegerter-Wilmsen et al., 2007*) and there is today accumulating evidence that many molecular actors involved in development are under both biochemical and mechanical control. In particular, studies on single animal cells show that mechanical forces contribute to the control of cell division and cell polarity, and relevant mechanotransduction pathways have been identified (e.g. *Houk et al., 2012*; *Fink et al., 2011*; *Thery et al., 2007*). The idea that mechanical stress may also play a role in defining cell identity has also emerged in the past decade (*Engler et al., 2006 Aliee et al., 2012*; *Landsberg et al., 2009*; *Farge, 2003*; *Brunet et al., 2013*). However, this

**eLife digest** The bending, stretching or squashing of cells or tissues can be used as a signal to trigger a range of biological responses. However investigating the role of these mechanical signals remains a challenge. This is partly because the forces that trigger the mechanical signals are often short-lived and changeable, and partly because the signals can be difficult to separate from the biochemical responses that they generate.

Stem cells present at the tip of the growing shoots in a plant are exposed to mechanical forces. These growing tips are called shoot meristems, and the stem cells they contain create all the aboveground organs of the plant (stems, leaves and flowers etc.). In each meristem, a boundary forms between the slow-growing stem cells at its centre and the fast-growing organs that form around them. Because these plant cells are both stuck together by their cell walls and growing at different rates, strong mechanical stresses are created causing this boundary to fold. A key regulator of the meristem is a protein called STM, but it remains unclear whether mechanical signals are involved in the control of this protein.

To investigate this, Landrein et al. tracked where the gene for the STM protein was switch on in shoot meristems in a plant called Arabidopsis, and found that it is highly active at the boundary. Analysing STM in different mutant plants combined with advanced imaging techniques revealed that STM activity correlates with the extent of creasing at this boundary. The STM protein is also required at the boundary to ensure that developing organs separate out. These findings suggest that boundary folding might somehow create signals that activate STM. One candidate signal was the plant hormone called auxin because reduced levels of this hormone were previously associated with boundary formation. However, in further experiments, Landrein et al. ruled out auxin's involvement in this process.

So do mechanical signals activate STM at the boundary? To test this, the mechanical forces in the meristem were altered by compressing the growing shoot meristems in miniature vices and by killing a few stem cells at the meristem centre. Both of these actions triggered the production of STM in the meristem, consistent with its activity being altered by mechanical stress.

Landrein et al. propose that the mechanical regulation identified acts in parallel to auxin signalling, providing robustness to the regulation of gene activity in the shoot meristem. In other words, tissue folding can guide gene expression, via the production of mechanical signals. But how shoot meristem cells respond at a molecular level to mechanical stress awaits future work. Finally, proteins related to STM can be found in all biological kingdoms, including some proteins that regulate important process in animal development. Whether the activity of these related proteins is also regulated by mechanical forces remains to be investigated.

pioneering work is still debated and a role of mechanical signals in cell identity during development remains an open question. Furthermore, all the currently known mechanotransduction pathways involve elements of established biochemical-based transduction pathway (e.g. *Janmey et al., 2013*; *Jaalouk and Lammerding, 2009*; *Orr et al., 2006*). Assuming mechanical signals play an important role in development, one may thus question the added value of mechanical signals in development if their action is so tightly coupled to biochemical signaling.

Because their development is slow, iterative and does not involve cell movements, plants are systems of choice to explore the possible contribution of mechanical signals in proprioception, i.e. in channeling growth and identity from intrinsically generated mechanical stresses. Mechanical cues contribute to the emergence of lateral roots in *Arabidopsis* (*Ditengou et al., 2008*; *Richter et al., 2009*), the expression level of the transcription factor *PtaZFP2* correlates to the amount of bending in poplar stems (*Coutand et al., 2009*) and the expression of the *ELA1* gene in a specific cell layer of the developing seed is triggered by mechanical signals that are generated by the growth of the embryo against the endosperm (*Creff et al., 2015*). While these studies suggest that physical forces can contribute to cell identity definition in plants, this remains to be fully demonstrated. A fitting system for this question is the shoot apical meristem (SAM), which contains a plant stem cell niche and controls the formation and identity of all aerial organs. Because cells are glued to each other, differential growth generates mechanical conflicts leading to shape changes. For instance, tissue folding

occurs in the boundary domain of the SAM between the slow growing meristem and the fast growing organ. Because the epidermis is load-bearing (**Dumais and Steele, 2000**; **Kutschera and Niklas, 2007**), the boundary domain is characterized by a highly anisotropic mechanical stress; this mechanical stress controls microtubule orientation, which in turn channels growth direction and promotes tissue folding (**Burian et al., 2013**; **Hamant et al., 2008**). A contribution of mechanical stress in the polarity of the auxin efflux carrier PIN-FORMED 1 (PIN1) was also proposed suggesting an indirect contribution of mechanical signals in auxin patterns and thus in organogenesis at the shoot apex (**Heisler et al., 2010**; **Nakayama et al., 2012**). While the genetic bases of meristem functions are now well documented, a link between these genetic regulators and mechanical signals remains to be identified in the SAM.

The homeodomain transcription factor SHOOT MERISTEMLESS (STM) is a key regulator of meristem functions and its expression is often considered as the best marker of meristematic identity. In the shoot apical meristem, *STM* is expressed ubiquitously, with the exception of young primordia where it is down-regulated (**Long et al., 1996**). Interestingly, the *STM* promoter has previously been reported to be more active in the boundary domains of the SAM (**Heisler et al., 2005**; **Leasure et al., 2009**; **Kim et al., 2003**; **Jurkuta et al., 2009**). In line with this observation, *STM* expression is regulated by several boundary specific genes (**Aida et al., 1999**; **Borghi et al., 2007**; **Lee et al., 2009**; **Takada et al., 2001**), in part *via* auxin signaling (**Aida et al., 2002**; **Treml et al., 2005**). Because the boundary domain is also a site under mechanical stress, we investigated whether such stresses could act as signals to control *STM* expression at the shoot apical meristem.

## Results

### *STM* promoter activity is quantitatively correlated to curvature at the meristem boundary

Because it is easily accessible, we focus our analysis on the shoot apical meristem (SAM) at inflorescence stage; the generated organs are thus floral meristems. We generated a transcriptional fusion *pSTM::CFP-N7* with the 5,7 kb region upstream of the *STM* gene (AT1G62360) and observed the presence of an enhanced *STM* promoter activity in the SAM boundaries, as previously shown (**Heisler et al., 2005**; **Leasure et al., 2009**; **Kim et al., 2003**; **Jurkuta et al., 2009**), n > 20, **Figure 1A**). To further confirm this result, which was obtained in dissected meristems from greenhouse-grown plants, we also analyzed the CFP signal in meristems from NPA-treated in vitro grown seedlings. In these conditions, polar auxin transport is inhibited and naked meristems are generated (**Grandjean et al., 2004**). When plants were taken off the drug and started to initiate new organs, higher CFP signal was observed specifically in the boundary domain (**Figure 1—figure supplement 1**).

When performing live-imaging of the meristem expressing *pSTM::CFP-N7*, we found a correlation between *STM* promoter activity in the boundaries and the progressive formation of a crease in this domain, at least qualitatively (**Figure 1B**). To assess this quantitatively, we measured the Gaussian curvature of the boundary at different stages of development in parallel with CFP signal intensity. Existing quantification protocols were not adequate: the replica method for instance involves the application of dental resin to make a cast of the meristem (e.g. **Kwiatkowska and Dumais, 2003**) and this may impact gene expression; the image analysis software *MorphoGraphX* could provide a mesh surface together with gene expression levels, but some errors in the most curvy parts of the meristem could not be avoided. We thus implemented the level set method (**Sethian, 1999**) in Python to detect the exact surface of the meristem: a suitable surface obtained by the thresholding of the image is evolved to the smooth surface that is the most accurate representation of the surface contrast in the 3D images. Second, *MorphoGraphX* was used to mesh the surface (**Kierzkowski et al., 2012**), and to compute the Gaussian curvature taking into account neighborhoods within a 15 μm radius (**Figures 1C** and **Figure 1—figure supplement 2**). Using this pipeline, a negative correlation between CFP signal intensity and Gaussian curvature could be revealed and quantified (**Figure 1C–E**, **Figure 1—figure supplement 2**). As a negative control, and using the same pipeline, no clear correlation between Gaussian curvature and CFP signal could be observed in a line expressing a transcriptional fusion *pPDF1::CFP-N7* that exhibits a relatively homogeneous signal in the whole meristem epidermis (**Figure 1—figure supplement 3**).

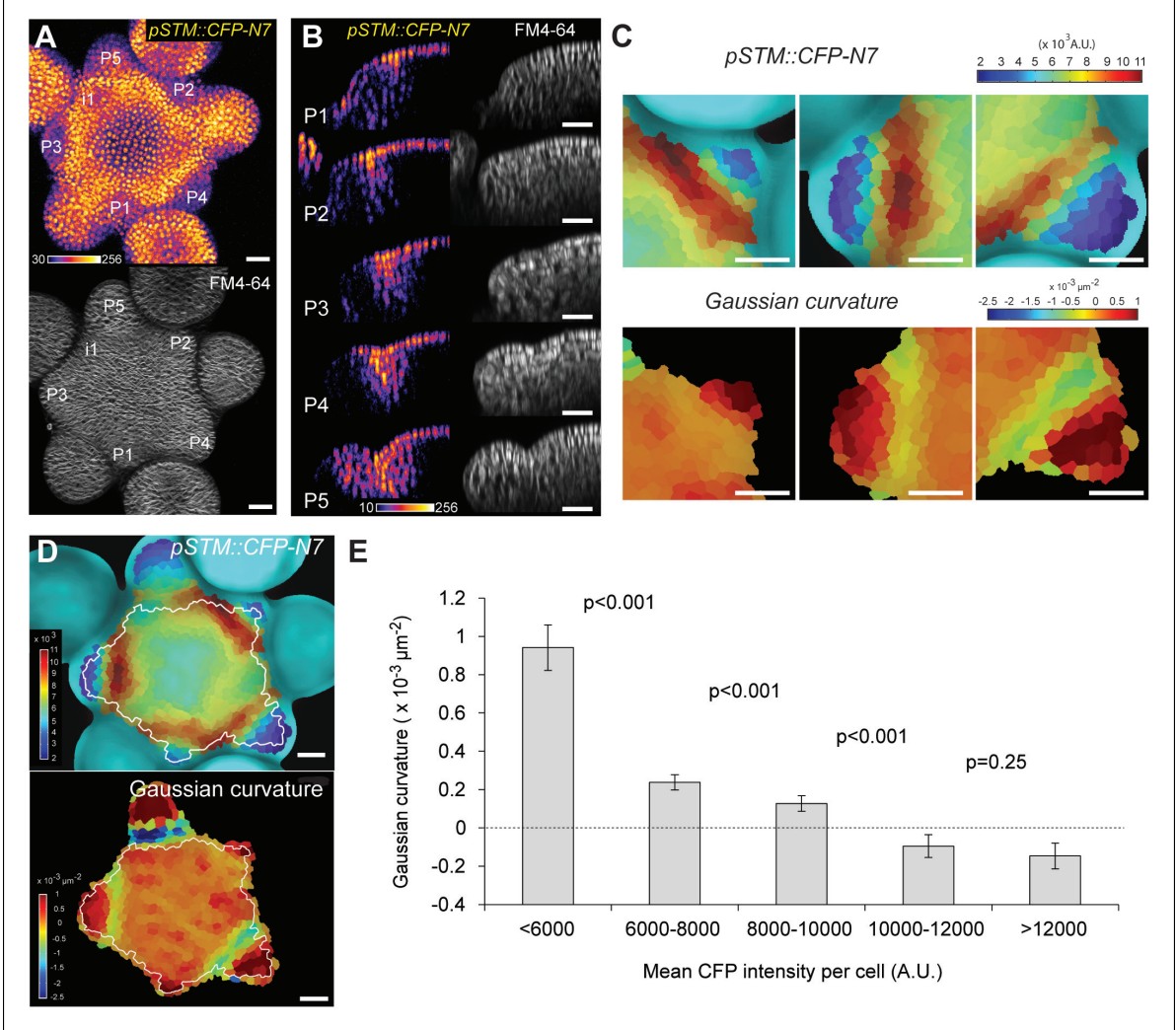

**Figure 1.** Correlation between *pSTM::CFP-N7* expression level and tissue folding at the boundary. (A) *pSTM::CFP-N7* expression pattern in the SAM. Membranes are labeled with FM4-64 (white, lower panel) and *pSTM::CFP-N7* expression is shown using the Fire lookup table in ImageJ (upper panel, n > 20). (B) Longitudinal optical sections (5 μm thick maximal projection of orthogonal views) through the middle of five successive boundaries of a representative meristem expressing *pSTM::CFP-N7*. Note the increase of *pSTM::CFP-N7* signal intensity in the boundary as the crease between organ and meristem becomes deeper. (C) Close-ups showing a correlation between *pSTM::CFP-N7* signal intensity (upper panels) and Gaussian curvature (lower panels, see Material and methods) in three successive boundaries of the meristem presented in A. (D and E) Quantification of the correlation between *pSTM::CFP-N7* signal intensity (upper panel) and Gaussian curvature (lower panel) in the meristem presented in A (see Material and methods). (D) The white outline encloses the cells that are used for the graph presented in E. (E) *pSTM::CFP-N7* signal intensity is plotted against Gaussian curvature. Values are compared using a bilateral Student test. The same correlation was observed in 5 independent meristems. Scale bars, 20 μm.

The following figure supplements are available for Figure 1:

**Figure supplement 1.** Time lapse imaging of a meristem recovering from NPA treatment and expressing *pSTM::CFP-N7*.

**Figure supplement 2.** Correlation between *pSTM::CFP-N7* expression level and tissue folding at the boundary.

**Figure supplement 3.** *pPDF1::CFP-N7* expression pattern in the SAM.

To test this correlation further, we also analyzed the expression pattern of *STM* in the *pSTM::ALcR AlcA::GFP* line (named hereafter *pBOUND>>GFP*) which contains 4.4 kb sequence upstream of the *STM* gene and has already been described previously (*Laufs et al., 2004*). In this line, expression is restricted to boundaries, thus confirming the correlation between the GFP signal intensity

and the extent of curvature at the boundary at least qualitatively (n > 20, *Figure 2A*). The correlation could also be observed in meristems from NPA-grown seedlings (*Figure 2—figure supplement 1*). In this line, the signal was so clear-cut, that the total area of GFP expressing cells on longitudinal sections could be qualitatively correlated with the formation of a crease (*Figure 2B*). To quantify this, we generated optical longitudinal section through the middle of each emerging primordia, measured the surface area of expression of *pBOUND>>GFP* in these sections and plotted it against the angle of tissue folding at the same position (see Material and methods). As in the *pSTM::CFP-N7* line, we measured a strong correlation between the area of *pBOUND>>GFP* expression and the angle of boundary folding measured on the orthogonal sections (n = 154, 5 SAMs observed at 5 time points over 48 hr, *Figure 2D*). A similar correlation could be measured in another independent live-imaging experiment (*Figure 2—figure supplement 2*, n = 193, 5 SAMs observed at 6 time points over 39 h).

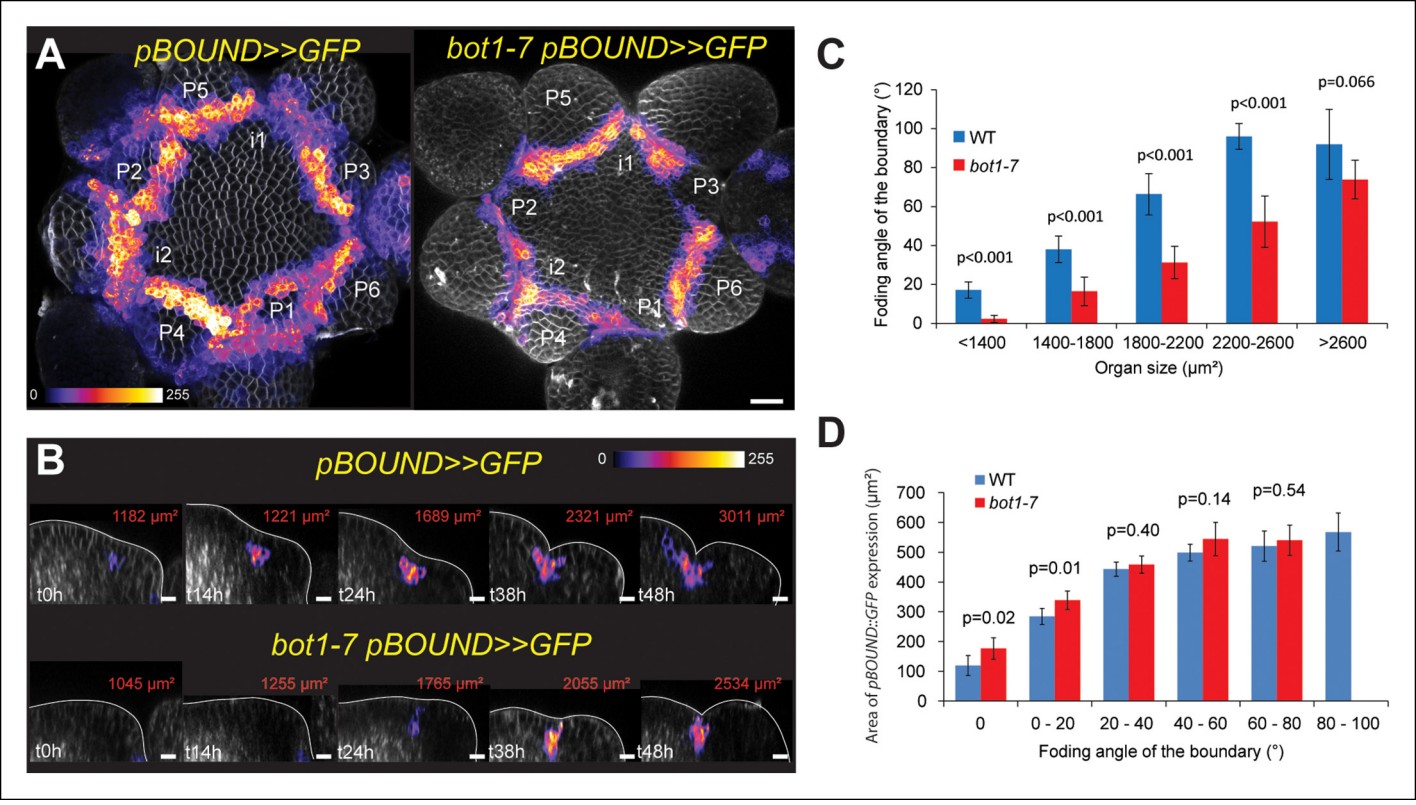

**Figure 2.** Correlation between *pBOUND>>GFP* expression level and tissue folding at the boundary in WT and *bot1-7*. (A) *pBOUND>>GFP* expression pattern in WT (ecotype *WS-4*) and *bot1-7* meristems. Membranes are labeled with FM4-64 (white) and *pBOUND>>GFP* expression is shown using the Fire lookup table in ImageJ. (B) Longitudinal sections through the middle of successive boundaries of the meristems presented in A (2 µm thick maximal projection of orthogonal views). Organ size (surface area as viewed from the top) is written in red for each stage. Note the delay in tissue folding and GFP signal expression in *bot1-7* when compared to the WT. The white line marks the outer surface of the SAM. (C) Quantification of the delay in curvature at the boundary in *bot1-7*: Folding angle is measured on orthogonal views and organ size is estimated from the measurement of surface area on top views. (D) The correlation between the folding angle of the boundary and the area of *pBOUND>>GFP* expression is maintained in *bot1-7* (both parameters are measured on orthogonal sections; WT: n = 130 from 5 SAM followed during a time lapse of 5 time points during 48 h, *bot1-7*: n = 79 from 3 SAM followed during a time lapse of 5 time points during 48 h). Values are displayed with a Student confidence interval (α = 0.05) and compared using a bilateral Student test. Scale bars: 20 µm.

The following figure supplements are available for Figure 2:

**Figure supplement 1.** Time lapse of a *pBOUND>>GFP* meristem recovering from NPA treatment.

**Figure supplement 2.** Correlation between pBOUND>>GFP expression area and tissue folding at the boundary.

To test the strength of the correlation between *STM* promoter activity and meristem shape, we next analyzed *STM* promoter activity in the *katanin* mutant allele *bot1-7* in which the presence of disorganized microtubules leads to the formation of a shallow crease at the boundary (*Uyttewaal et al., 2012*; *Bichet et al., 2001*). We reasoned that if *STM* promoter activity is truly correlated with curvature at the boundary, we should still be able to detect such correlation in the *bot1-7* mutant background, although with a delay in time. First, we measured the folding angle with respect to organ size in *bot1-7*. As expected, we observed a reduction in folding angle for a given primordium size in *bot1-7* when compared to the WT, demonstrating that the relation between tissue folding and organ emergence is impaired in *bot1-7*: large organs still have shallow boundaries (*Figure 2A,B,C*). Next, we measured the GFP signal area in the boundaries of the *bot1-7 pBOUND>>GFP* line as described above. Note that both *bot1-7* and *pBOUND>>GFP* are in the WS-4 ecotype, allowing comparison between mutant and WT backgrounds. Whereas *pBOUND>>GFP* expression area was still correlated to tissue folding, we also observed a reduction in the area of GFP expression for a given primordium stage, when compared to the WT indicating a delay in *pBOUND>>GFP* appearance in *bot1-7* (*Figure 2D*, WT: n = 154 measured on 5 meristems at 5 time points; *bot1-7*: n = 79 measured on 4 meristems at 5 time points). Therefore, modifying the shape of the SAM in *bot1-7* did not abolish the correlation between *STM* promoter activity and curvature, and instead demonstrated that *STM* promoter activity scales to Gaussian curvature values even when the relation between organ size and boundary shape is affected. Based on these results, *STM* promoter activity can thus be considered as a read-out of the extent of folding in the boundary of the SAM.

## *STM* expression at the boundary is required for organ separation

Even though we find a correlation between *STM* expression and tissue folding, it is not clear whether *STM* promoter activity at the boundary using a reporter line truly recapitulates *STM* expression in the SAM.

To check this, we first crossed the strong *stm* mutant allele *stm-dgh6* (*Aida et al., 2002*) with a previously described *pSTM::STM-Venus* line, whose expression is enhanced in boundaries, as observed in our transcriptional marker lines (*Heisler et al., 2005*; *Besnard et al., 2014; Figure 3A*). The *stm-dgh6* mutant exhibits the typical *stm* phenotype with two partly fused cotyledons, at an angle divergent from 180°, and a development arrest. At the same stage, the WT can exhibit up to 8 leaves while the *stm-dgh6* only displays two old cotyledons (*Figure 3B*). As observed in other strong *stm* alleles, *stm-dgh6* was able in rare cases to reinitiate organogenesis and generate a lot of vegetative tissues. In our growth conditions, we found that roughly 1out of 30 *stm-dgh6* plants managed to grow beyond the cotyledon stage: these plants generated many whorled leaves and a few sterile flowers (*Figure 3C*). To test whether *pSTM::STM-Venus* can complement the *stm-dgh6* phenotype, we selected 96 plants homozygous for the *pSTM::STM-Venus* construct and segregating the *stm-dgh6* mutation. After genotyping for the presence of the *stm-dgh6* mutation, we could not distinguish the WT, *stm-dgh6* heterozygotes and *stm-dgh6* homozygote plants visually, confirming that *pSTM::STM-Venus* can fully complement the mutation (*Figure 3D-F*, *Figure 3—figure supplement 1*). Importantly, we checked that STM-Venus was expressed in the *stm-dgh6* homozygote background by confocal microscopy. We found that the signal was comparable to that of *pSTM::STM-Venus* in a WT background, notably with an enhanced signal in boundaries (*Figure 3G*). In a few plants, we also observed larger stems and meristems, suggesting that *pSTM::STM-Venus* expression level might be slightly higher than that of the endogenous gene. These results demonstrate that higher *STM* promoter activity at the boundary in the marker lines reflects the endogenous *STM* expression pattern. Based on these data, and to ease the reading for the rest of the paper, we will use the wording '*STM* expression' when analyzing the fluorescent signal in *STM* transcriptional reporter lines.

To further confirm the biological relevance of higher *STM* promoter activity at the boundary, we next attempted to down-regulate *STM* expression at the boundary. Genetic evidence suggests that, in addition to its role in meristem maintenance, *STM* also contributes to organ separation in the meristem, in conjunction with other boundary expressed genes, such as *CUC*, *KNAT6*, *JLO* and *LOF* genes (*Lee et al., 2009*; *Aida et al., 2002*; *Belles-Boix et al., 2006*; *Spinelli et al., 2011*; *Borghi et al., 2007*; *Aida et al., 1999*). Lack of STM activity can generate organ fusions (e.g.

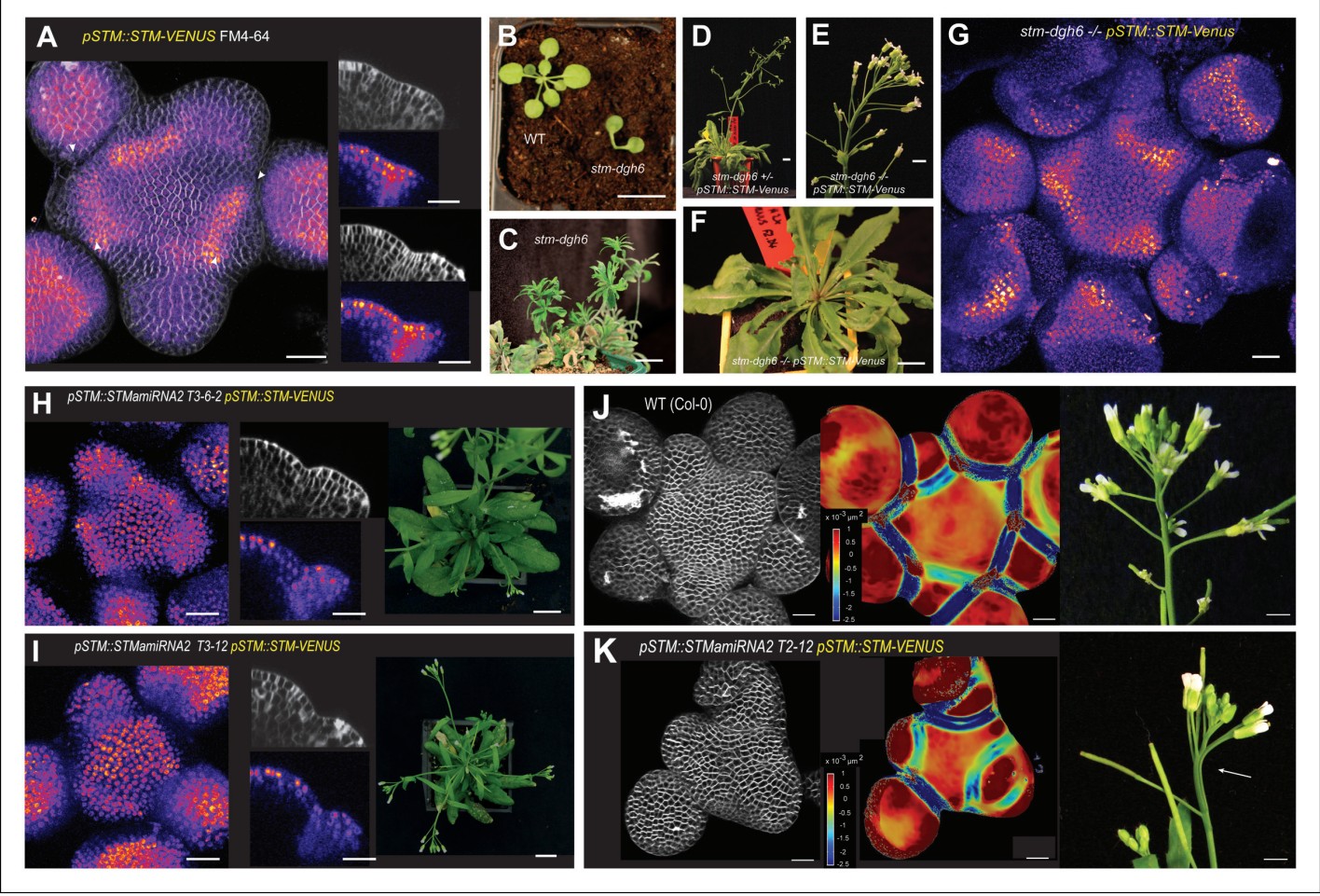

**Figure 3.** Organ separation requires *STM* expression at the boundary. (**A**) Representative expression pattern of the translational fusion *pSTM::STM-Venus* in a FM4-64 stained meristem showing an increased signal intensity in boundaries. Scale bar, 20 μm. (**B–F**) The translational fusion *pSTM::STM-Venus* partially rescues the phenotype of the strong mutant allele *stm-dgh6* (n = 5). (**B**) Phenotype of 3-week-old WT and *stm-dgh6* plants. Note the absence of postembryonic organs in the mutant. Scale bars, 1 cm. Aerial phenotype of 2-month old *stm-dgh6* plants. Scale bar, 1 cm. (**D**) Representative *stm-dgh6 pSTM::STM-Venus* plant. Scale bar, 1 cm. (**E**) Representative *stm-dgh6 pSTM::STM-Venus* inflorescence. Scale bar, 1 cm. (**F**) Representative *stm-dgh6 pSTM::STM-Venus* rosette. Scale bar, 1 cm. (**G**) Representative expression pattern of the translational fusion *pSTM::STM-Venus* in a *stm-dgh6* (-/-) meristem showing a similar expression pattern as in the WT. Scale bar, 20 μm. (**H, I**) Homogeneous expression pattern of the translational fusion *pSTM::STM-Venus* in two independent *pSTM::STMamiRNA* lines. Scale bar (microscopy), 20 μm. Scale bar (whole plant), 1 cm. (**J**) FM4-64 stained WT meristem (ecotype *Col-0*), Gaussian curvature extracted using the level set method and MorphoGraphX, Scale bar (microscopy), 20 μm. Scale bar (inflorescences), 1 cm. (**K**) FM4-64 stained *pSTM::STMamiRNA pSTM::STM-Venus* meristem, Gaussian curvature extracted using the level set method and MorphoGraphX. Scale bar, 20 μm. Boundaries do not scale to the reduced meristem size; inflorescence phenotype with fusion events. Scale bar, 1 cm.

The following figure supplements are available for Figure 3:

**Figure supplement 1.** Molecular characterization of the *stm-dgh6 pSTM::STM-Venus.*

**Figure supplement 2.** Molecular characterization of the *pSTM::STMamiRNA* lines.

*Endrizzi et al., 1996*) and can enhance the fusion phenotype of *cuc1* and *cuc2* mutant seedlings, revealing a role of *STM* in boundary formation during the early stages of embryo development (*Aida et al., 1999*; *2002*). This function also involves the *STM* paralog *KNAT6* gene, which is expressed in the boundary domain of the meristem in embryos (*Belles-Boix et al., 2006*). Based on this literature, the expression of *STM* at the boundary is generally thought to control organ separation, but this has not been formally demonstrated.

To address this hypothesis, we generated lines expressing an artificial microRNA (amiRNA) against *STM* under the control of the *STM* promoter. In these lines, *STM* expression level was not always negatively correlated to *STM* amiRNA expression level; yet, *STM* expression was down-regulated in all the lines we analyzed showing that this amiRNA can effectively inactivate *STM* to some extent (*Figure 3—figure supplement 2*). Given the expression pattern in the *STM* transcriptional marker lines, we reasoned that in these lines, we should primarily inactivate *STM* in the boundaries and to a lesser extent in the rest of the meristem. To test whether the amiRNA was effective at downregulating *STM* mRNA accumulation in boundaries, we introgressed the *pSTM::STMamiRNA* in the *pSTM::STM-Venus* lines. As expected, the STM-Venus signal was slightly (*Figure 3H*) or more strongly reduced (*Figure 3I*) at the boundary, leading to a homogeneous STM-Venus signal in the meristem. Although meristem size was affected in these lines (*Figure 3J and K*), meristem termination defects were rare when compared to *stm* mutants. In particular, many normal and fertile flowers were generated (*Figure 3K*). We also observed that Gaussian curvature in the boundaries did not scale to meristem size (*Figure 3K*). Strikingly, major fusion defects were observed in most of the lines we generated, thus showing that organ separation was impaired in these lines (*Figure 3K*). The presence of wide boundaries (relative to meristem size) together with a reduction in meristem size that brings adjacent boundaries next to one another, may cause such fusion events. Because the fusions are revealed much later in development, we cannot completely exclude other scenarios. Note that *STM* mRNA, protein or amiRNA can move between non-boundary cells and boundary cells. Yet, as a negative impact of the amiRNA on STM-Venus signal could be detected at the boundary, these data rather confirm that *STM* expression at the boundary has functional implications and further support its role in organ separation.

## Auxin depletion is not absolutely required for *STM* expression at the boundary

The fact that *STM* expression correlates with curvature at the boundary can simply be explained by the fact that STM reduces the growth rate at the boundary, leading to tissue folding. *KNOX* genes can cause alterations in curvature in otherwise smooth surface (*Long and Barton, 1998*; *Barkoulas et al., 2008*). The impact of STM on crease formation at the boundary could be mediated by inhibiting cell growth in this domain. Consistent with this scenario, *KNOX* target genes include genes involved in auxin transport as well as genes involved in cell wall synthesis (*Bolduc et al., 2012*). Given the strong correlation between *STM* promoter activity and tissue folding, we investigated whether a signal related to curvature could add robustness to the *STM* expression pattern at the boundary.

The plant hormone auxin could play such a role. A local auxin peak is one of the earliest marker of organ initiation, and conversely auxin depletion is an early marker of the boundary domain (*de Reuille et al., 2006*; *Reinhardt et al., 2003*; *Heisler et al., 2005*). As auxin keeps accumulating in the organ, outgrowth goes on and curvature at the boundary increases. Interestingly, auxin and KNOX proteins are known to act antagonistically. Disrupting auxin transport with NPA or in *pin1* and *pid* mutant enhances class I *KNOX* gene expression (*Scanlon, 2003*; *Furutani et al., 2004*). Conversely, *KNAT1* inactivation can partially rescue the leaf number defects in *pin1*, suggesting that the auxin signaling pathway also promotes organ emergence by repressing the expression of *KNOX* genes (*Hay et al., 2006*). In the boundary, the PIN1-dependent auxin minimum was also shown to promote axillary meristem formation whereas ectopic auxin production in the boundary inhibits axillary meristem formation (*Wang et al., 2014*). This is further consolidated by the analysis of the DII-Venus auxin sensor, which is degraded in the presence of auxin and exhibits a strong signal at the boundary (*Brunoud et al., 2012*; *Vernoux et al., 2011*). Consistent with the characteristic localization of *PIN1* transporters around the boundary (*Figure 4—figure supplement 1; Heisler et al., 2005*), we also observed that a higher DII-Venus signal in the boundaries correlates with an increase in *pSTM::CFP-N7* expression (*Figure 4A*). Interestingly, while this correlation could also be observed on longitudinal sections through the boundary as it folds (*Figure 4B*), we also found that this correlation is stronger in the L1 layer: In the L2 layer, *pSTM::CFP-N7* expression increased during crease formation, but this was not always correlated with an increase in DII-Venus signal (*Figure 4B*).

So far, our data are thus consistent with a scenario in which a local depletion of auxin leads to the initial induction of *STM* expression at the boundary. If this scenario were true, the application of exogenous auxin should thus inhibit *STM* expression at the boundary. To test this hypothesis, we

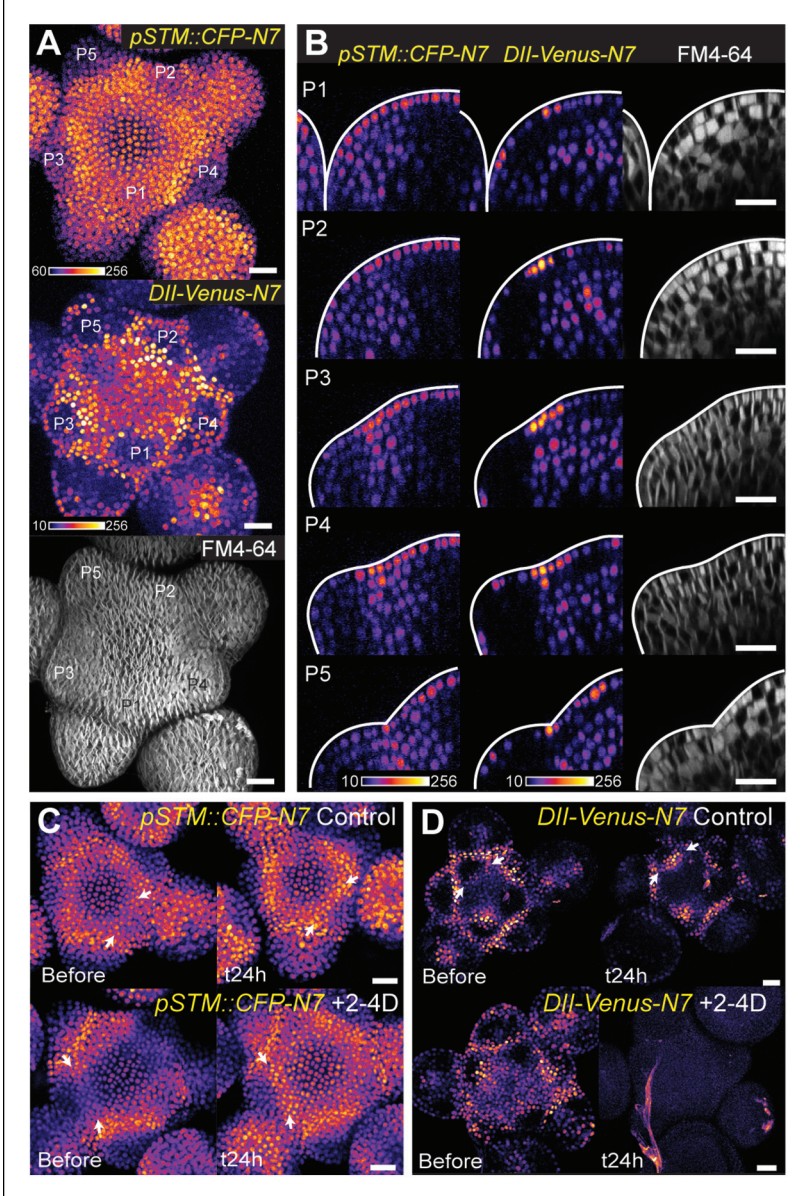

**Figure 4.** The DII-Venus and *pSTM::CFP-N7* signals largely overlap and can be uncoupled. (**A**) Projection of a representative meristem expressing both *pSTM::CFP-N7* and *DII-Venus-N7* and stained with FM4-64: both signals are induced in the boundary. (**B**) Orthogonal sections through the middle of the boundaries of the successive primordia of the SAM presented in showing an overlap of both signals, except in the L2 layer. (**C, D**) Overnight treatment with 10 μM of synthetic auxin 2,4-D on dissected meristems: (**C**) no effect on *pSTM-CFP* expression after 2,4-D application (Control: n = 3, 2-4D treatment: n = 3). White arrows point at new *CFP* signals in boundaries (**C** and **D**); (**D**) total degradation of *DII-Venus* after 2,4-D application (Control: n = 11, 2-4D treatment: n = 12). Scale bars, 20 μm.

The following figure supplements are available for Figure 4:

**Figure supplement 1.** PIN1 localization in the SAM.

applied the synthetic auxin 2,4-D onto the meristem and observed the expression of *STM*. As expected, DII-Venus levels were dramatically decreased in these conditions ([*Brunoud et al., 2012*]; *Figure 4D*), confirming that auxin could indeed diffuse into the meristem. In the same conditions, *STM* expression at the boundary was however either unchanged or even slightly increased

(*Figure 4C*). This suggests that, whereas auxin levels negatively correlate with *STM* expression in the meristem, this correlation is also dispensable: *STM* expression and auxin depletion can be uncoupled at the boundary. This also shows that the relation between auxin and *STM* expression is more complex. Next we investigated whether the involvement of mechanical signals in *STM* expression at the boundary could help clarify these discrepancies.

## *STM* expression is induced by mechanical stress

There is growing evidence that mechanical stresses act as instructive signals in parallel to biochemical signals in development. These stresses have notably been involved in the progression of gastrulation in Drosophila embryos (e.g. *Farge, 2003*; *Lecuit and Lenne, 2007*) or in the formation of leaves with ruffled edges (*Audoly and Boudaoud, 2003*; *Nath et al., 2003*). In the meristem, the boundary separates a fast growing organ from the slow growing meristem. Mechanical stresses thus emerge early on from differential growth in the boundary and have been shown to act as instructive signals controlling the behavior of cortical microtubules and PIN1 early on. In turn, this promotes tissue folding that further reinforces the stress pattern at the boundary (*Hamant et al., 2008*; *Heisler et al., 2010*). We thus explored a scenario in which mechanical stress may be sufficient to induce and reinforce *STM* expression at the boundary. To do so, we investigated whether mechanical perturbations can trigger *STM* expression.

Note that mechanical perturbations can be of different nature. Mechanical perturbations induced by wind or touch, are extrinsic and discontinuous; they cause short-term elastic deformations (e.g. transient bending). They can impact gene expression within minutes and induce major developmental responses such as stem thickening, when repeated (*Braam and Davis, 1990*). In the context of meristem growth, mechanical (tensile) stresses are in contrast intrinsic and continuous, and they cause long-term plastic responses (growth). The origin of such growth-related stresses lies in the presence of high turgor pressure rather than external stimuli. The opposition between these two kind of stresses can be illustrated with the microtubule response: transient pinching does not lead to supracellular microtubule alignments in the meristem, and this has even served as a negative control for the microtubule response to ablations in this tissue (*Hamant et al., 2008*). Nonetheless we cannot exclude the possibility that some elements of the mechanotransduction pathways are shared between these two types of mechanical signals. Here, given our correlation between *STM* expression and crease formation at the boundary, we designed our mechanical tests to check whether *STM* expression can be modified by continuous mechanical perturbations.

First, we modified the mechanical stress pattern in the SAM using compressions or ablations and we followed the impact on the *pSTM::CFP-N7* and *pBOUND>>GFP* expression patterns. Note that in all these experiments, we used in vitro-grown plants with a naked meristem recovering from a NPA treatment, as plants from the greenhouse need to be dissected to access the SAM, and the wounds may interfere with the mechanical perturbations. Using a microvice, we first induced global compression of the SAM. This is predicted to increase tension in the epidermis, and in the most extreme cases, to induce a bias in maximal stress direction parallel to the blades (*Figure 5A*, [*Hamant et al., 2008*; *Uyttewaal et al., 2012*]). Around 8 h after compression, we detected an increase of *pSTM::CFP-N7* signal (n = 8; *Figure 5B* and *Video 1*). Similar induction was observed in the *pBOUND>>GFP* line; although the GFP signal was more variable before compression, a strong induction could nevertheless be observed in the *pBOUND>>GFP* line after compression (n = 11; *Figure 5C* and *Video 2*). Furthermore, even if the induction of *STM* expression occurred earlier in the *pSTM::CFP-N7* line than in the *pBOUND>>GFP* line (*Figure 5B and C*), the overall time frame (8 h onwards) rather suggests an indirect effect of compressions on *STM* expression.

To check whether *STM* can also be induced after a more local mechanical perturbation, we next performed ablations of a few cells in the meristem to induce high circumferential stress around the site of ablation (*Figure 5D* [*Hamant et al., 2008*; *Heisler et al., 2010*; *Uyttewaal et al., 2012*]). Strikingly, we observed a local increase of *pSTM::CFP-N7* signal intensity around the ablation site from t = 18 h onwards (n = 16; *Figures 5E* and *Video 3*). The induction also seemed stronger in larger ablations than in smaller ablations, consistent with a greater range of stress perturbation in larger ablations (*Figure 5—figure supplement 1*). A similar response was also observed in the *pBOUND>>GFP* background, again with a delay (n = 12; *Figure 5F* and *Video 4*). While some degree of variability between individual plants at t = 0h could be observed, *pBOUND>>GFP* expression was also consistently induced after ablation (*Figure 5F*). We also noticed that with both

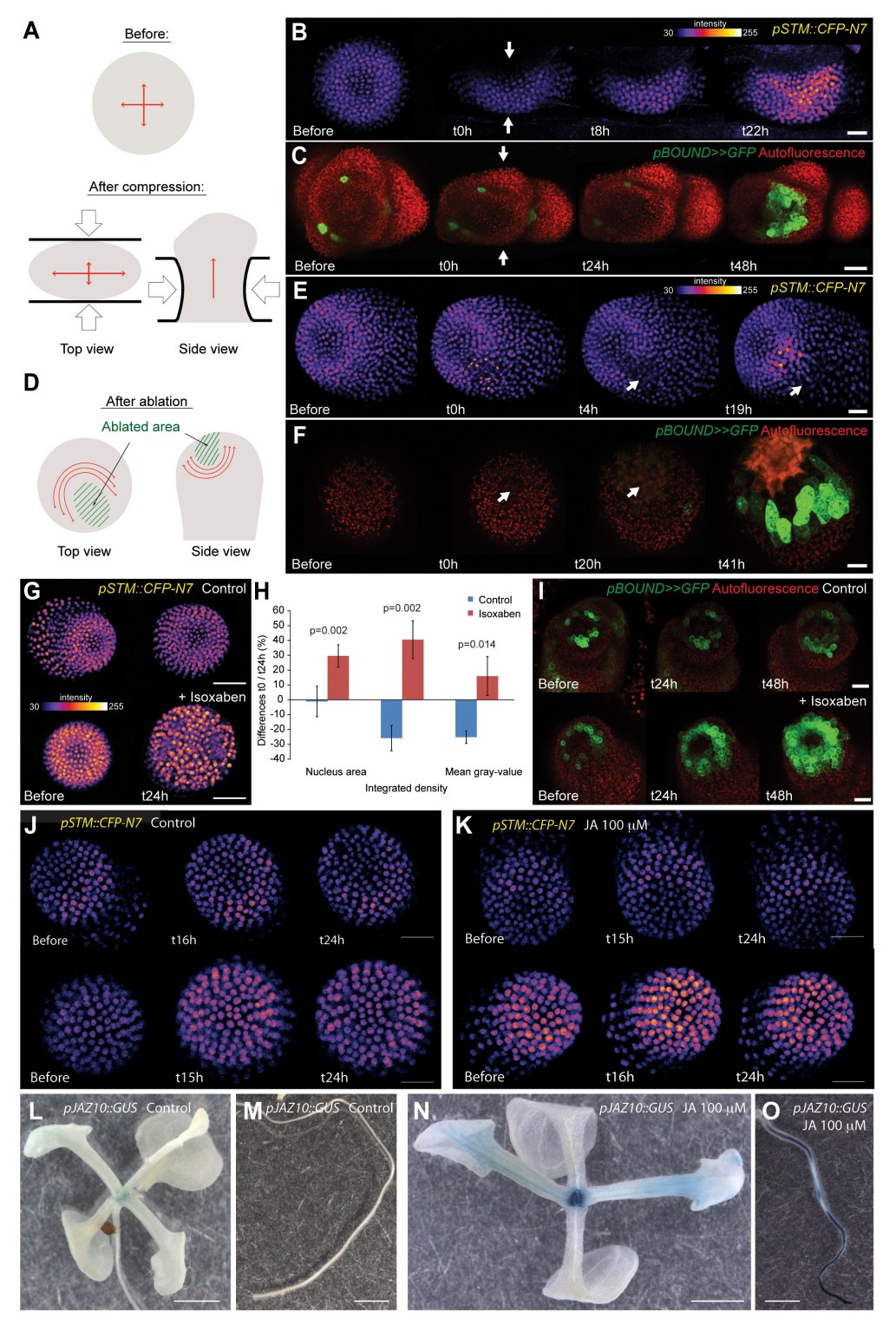

**Figure 5.** *STM* expression can be induced by mechanical perturbations. (**A to C**) Global compression of meristems with a microvice lead to an increase in *STM* expression (arrows indicate the direction of the compression). (**A**) Predicted impact of compression on the mechanical stress pattern. (**B**) *pSTM::CFP-N7* signal before and after compression in a representative meristem (n = 8). (**C**) *pBOUND>>GFP* signal before and after compression in a representative meristem (n = 11, red dots correspond to plast auto-fluorescence). (**D to F**) Ablation of a small number of cells leads to an increase in *STM* expression (white arrows indicate the site of ablation). (**D**) Predicted

*Figure 5. continued on next page*

*Figure 5. Continued*

impact of a local ablation on the mechanical stress pattern. (E) *pSTM::CFP-N7* signal before and after ablation with a needle (n > 30). (F) *pBOUND>>GFP* signal before and after ablation using a pulsed UV laser (n > 12, red dots correspond to plast auto-fluorescence). (G and H) Isoxaben treatment leads to an increase of *pSTM::CFP-N7* signal. (G) Representative *pSTM::CFP-N7* signal after overnight immersion in water with DMSO (upper panel) or in 10 µM isoxaben (lower panel). Note the increased nucleus size after isoxaben treatment, consistent with increased endoreduplication levels. (H) Quantifications: CFP signal intensity in 10 nuclei from the central zone of 6 isoxaben-treated meristems and 7 water-treated meristems. Values are displayed with a Student confidence interval ($\alpha$ = 0.05) and compared using a bilateral Student test. (I) Isoxaben treatment leads to an increase in *pBOUND>>GFP* signal (red dots correspond to plast auto-fluorescence). (I, left) Representative *pBOUND>>GFP* signal after overnight immersion in water (n = 10). (I, right) Representative *pBOUND>>GFP* signal after overnight immersion in 5 to 20 µM isoxaben (n = 20). Scale bars, 20 µm. (J, K) Jasmonate does not enhance *STM* promoter activity (J) *pSTM::CFP-N7* signal after prolonged incubation in water supplemented with 1/1000 V/V ethanol. (K) *pSTM::CFP-N7* signal after prolonged incubation in water supplemented with 100 µM jasmonate diluted in ethanol (1/1000 V/ V). Scale bars, 20 µm. (L-O) Jasmonate enhances *pJAZ10* promoter activity. (L, M) Aerial part (L) and root (M) of 3 week old NPA grown seedlings. *pJAZ10::GUS* staining after overnight incubation in water supplemented with 1/1000 V/V ethanol (n = 8). (N,O) Aerial part (N) and root (O) of 3 week old NPA grown seedlings. *pJAZ10::GUS* staining after overnight incubation in water supplemented with 100 µM jasmonate diluted in ethanol (1/1000 V/V) (n = 14). Scale bars, 0.5 cm.

The following figure supplements are available for Figure 5:

**Figure supplement 1.** *pSTM::CFP-N7* induction after ablations of different sizes.

**Figure supplement 2.** *STM* mRNA distribution after ablation in the SAM.

markers, the induction of *STM* was not homogeneous around the ablation site. In particular, *STM* was never induced at the base of meristem, that is in the differentiating cells from the upper part of the stem. This is consistent with the prevailing view from the literature stating that the competence to express *STM* is limited to meristematic cells *sensu stricto*, and our data suggest that such a pre-pattern cannot be overridden by mechanical perturbations.

To confirm these results, we analyzed the distribution of *STM* mRNA by whole mount in situ hybridizations on shoot apices after ablation (*Rozier et al., 2014*). Even though this method is only semi-quantitative, we could detect an asymmetric signal, with a higher intensity near the ablation site, consistent with the results obtained with the *pSTM::CFP-N7* and *pBOUND>>GFP* lines after ablation (*Figure 5—figure supplement 2*, n = 9). Note that ablation may also provide increased accessibility to the probe. Last, to check whether *STM* expression would be enhanced by wound-induced jasmonate production, we incubated shoot apices with 100 µM jasmonate over night. In these conditions, we did not observe any significant induction of *pSTM::CFP-N7* signal in most of the plants (N = 10/14, *Figure 5J*), and slight fluctuations in CFP signal in the remaining ones (N = 4/14, *Figure 5K*). Based on these results, we cannot rule out completely that jasmonate interferes with *STM* expression. Yet, as this contrasts with the systematic and steady induction of *STM* after ablation and with the robust induction of a *pJAZ10::GUS* reporter by jasmonate (*Figure 5L-O*), wound-induced jasmonate is not the most likely candidate as a secondary messenger between stress and *STM* induction.

To further confirm an induction of *STM* expression by mechanical signals, we next modified the mechanical stress level using isoxaben treatments. Isoxaben is a well-known inhibitor of cellulose synthesis; after treatment, walls become mechanically weaker, and thus tensile stress increases (*Heisler et al., 2010*; *Uyttewaal et al., 2012*). In these conditions, we could detect a global increase in *pSTM-CFP-N7* signal intensity, when compared to that of the control (measurement on 10 nucleus of the central zone in the control (n = 6 meristems) and isoxaben-treated (n = 7 meristems); *Figure 5G and H*). This response was also confirmed in the *pBOUND>>GFP* background (control: n = 10, isoxaben: n = 20; *Figure 5I*). Note that *pBOUND>>GFP* was not induced in the entire meristem after isoxaben treatment, further demonstrating that mechanical perturbations cannot override the prepattern in the SAM. Because these different experimental setups have little in common

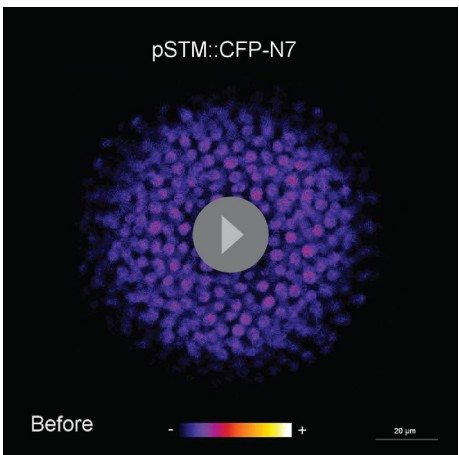

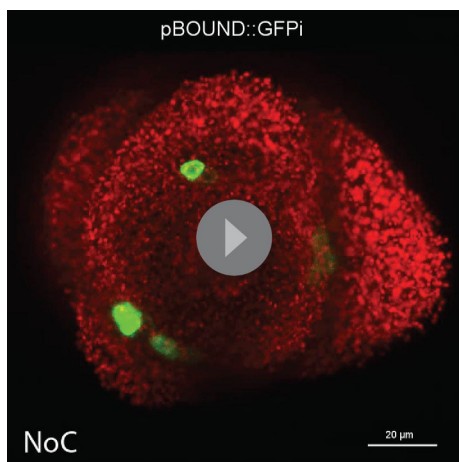

**Video 1.** *pSTM::CFP-N7* after compression (relates to *Figure 5B*).

**Video 2.** *pBOUND>>GFP* after compression (relates to *Figure 5C*).

except mechanical stress perturbation, we thus propose that *STM* expression can indeed be modulated by mechanical stress in the SAM.

## Boundary expressed genes are not systematically induced after mechanical perturbations

One may wonder whether such artificial mechanical perturbations could induce the expression of all meristematic genes, and in particular other boundary-expressed genes, as a non-specific stress response. To test this hypothesis, we first focused on *PINOID (PID)*. Like *STM* in the boundaries, *PID* has an established role in organ separation and boundary function (*Furutani et al., 2004*; *Reinhardt et al., 2003*). In a *pPID::PID-YFP* line, we observed a more abundant signal at the boundary of the meristem, as in the *pSTM::STM-Venus* line (*Figure 6A*). As detailed expression data were lacking in the meristem, we first analyzed the activity of the *PID* promoter. We generated a *pPID::CFP-N7* line and observed a pattern that somewhat echoes that of *pSTM::CFP-N7*. In particular, the CFP signal was detected in the entire meristem and was enhanced in the boundary domain (*Figure 6B*). The signal obtained in a *pPID::AlcR ALcA::GFP* line (*pPID>>GFP*) further confirmed this result, echoing the *pBOUND>>GFP* signal (*Figure 6C and D*). However, after ablation, *PID* expression was not affected in both the *pPID::CFP-N7* and *pPID>>GFP* lines, in contrast to our results in

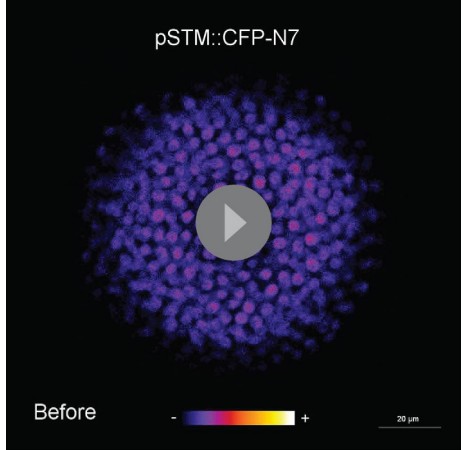

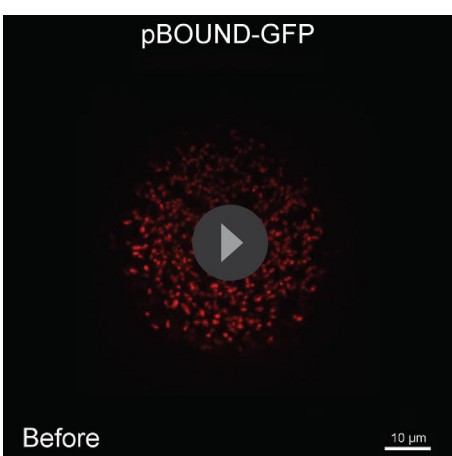

**Video 3.** *pSTM::CFP-N7* after an ablation (relates to *Figure 5E*).

**Video 4.** *pBOUND>>GFP* after an ablation (relates to *Figure 5F*).

the *pSTM::CFP-N7* and *pBOUND>>GFP* line (*Figure 6E–G*, *pPID::CFP-N7*: n = 13; *pPID>>GFP*: n = 7). No induction could also be detected after isoxaben treatment in the *pPID>>GFP* line (*Figure 6— figure supplement 1*; n = 11). Therefore, in our hands *PID* expression appeared as a negative control for our mechanical perturbations: *PID* expression at the boundary may not depend on a mechanical signal. Conversely, it suggests that the induction of *STM* by mechanical stress retains some specificity.

To further check whether other boundary expressed transcription factors are induced by mechanical perturbations, we performed ablations in *pCUC1::CUC1-GFP* and *pCUC3::CFP* expressing meristems. As ablations are also inducing other signals than mechanical signals, such a test is not sufficient to know whether a gene is induced by mechanical stress. Yet, induction after wounding is a necessary condition for a gene to respond to mechanical perturbation and thus this simple test can help to identify genes whose expression is insensitive to mechanical stress.

*CUC1* and *CUC3* belong to the NAC family of transcription factors and are two well-known regulators of boundary function at the shoot apical meristem (*Aida and Tasaka, 2006*). Note that in contrast to *CUC3* for which a transcriptional fusion is sufficient to recapitulate boundary expression, *CUC1* expression is restricted to the boundary *via* the post-transcriptional action of miR164, and thus strict boundary expression of *CUC1* can only be observed in a translational fusion line (*Sieber et al., 2007*; *Laufs et al., 2004*). After ablations, no significant induction could be observed in the *pCUC1::CUC1-GFP* line (*Figure 6—figure supplement 2*; n = 11). For instance, when comparing the time points before and 24 h after ablations, GFP signal intensities were roughly identical and the GFP pattern could not be easily related to the wound position or shape. Later on, as the meristem recovered from the wound, new boundaries were initiated in which new GFP signal could be observed, again with no relation to the ablation position or shape (*Figure 6—figure supplement 2*). Sometimes, the new CUC1-GFP signal even formed a line perpendicular to the wound edge, corresponding to the initiation of the boundary of following organs (*Figure 6—figure supplement 2*, bottom panels), in contrast to *pSTM::CFP-N7* and *pBOUND>>GFP* induction that always appear and further consolidates along the wound edge. In other words, the cycle of *CUC1* induction at the boundary went on largely undisturbed, independent of the presence of a neighboring ablation. Altogether, this strongly suggests that *CUC1* expression at the boundary is also not controlled by a mechanical signal.

The response of the *pCUC3::CFP* line to ablation was completely different than that of the *pCUC1::CUC1-GFP* line. In the *pCUC3::CFP* line, we observed a strong and steady induction of CFP signal after ablation in all the analyzed meristems (n = 20). Interestingly, this induction seemed to appear earlier than in the *pSTM::CFP-N7* line, as a strong induction could be detected as early as t = 6 hr after ablation (*Figure 6—figure supplement 3*). Although these data may suggest that *CUC3* expression is controlled by a mechanical signal, further tests would be required to reach such a conclusion. Nonetheless this demonstrates that boundary expressed transcription factors, even from the same family, are not systematically induced by wounding and, thus, that the mechanical induction of *STM* retains some specificity.

## *STM* induction after mechanical perturbations can be partly uncoupled from PIN1-dependent auxin distribution

Another issue raised by our mechanical perturbations is the relatively long time responses (8 to 24 hr) for *STM* to be induced. While this is consistent with the relatively slow timing of organ emergence and boundary folding in Arabidopsis meristems (*Kwiatkowska, 2004*; *Reddy et al., 2004*; *Grandjean et al., 2004*), it strongly suggests that the induction of *STM* expression after mechanical perturbations is indirect and could involve secondary signals, such as hormones. In line with this hypothesis, mechanical signals in animal development have all been shown to interfere with established biochemical signal transduction pathways (e.g. [*Janmey et al., 2013*; *Jaalouk and Lammerding, 2009*; *Orr et al., 2006*]). Auxin signaling could again play such a role at the boundary, as the polarity of the auxin efflux carrier PIN1 depends in part on membrane tension (*Heisler et al., 2010*; *Nakayama et al., 2012*). Based on these data, a scenario emerges where mechanical stress would control PIN1 localization at the boundary, which in turn would locally deplete this domain from auxin, leading to a local induction of *STM* expression. Although our previous results suggest that *STM* expression at the boundary does not solely rely on auxin depletion (based on a global auxin treatment, see *Figure 4C,D*), it does not formally exclude the possibility that the contribution of

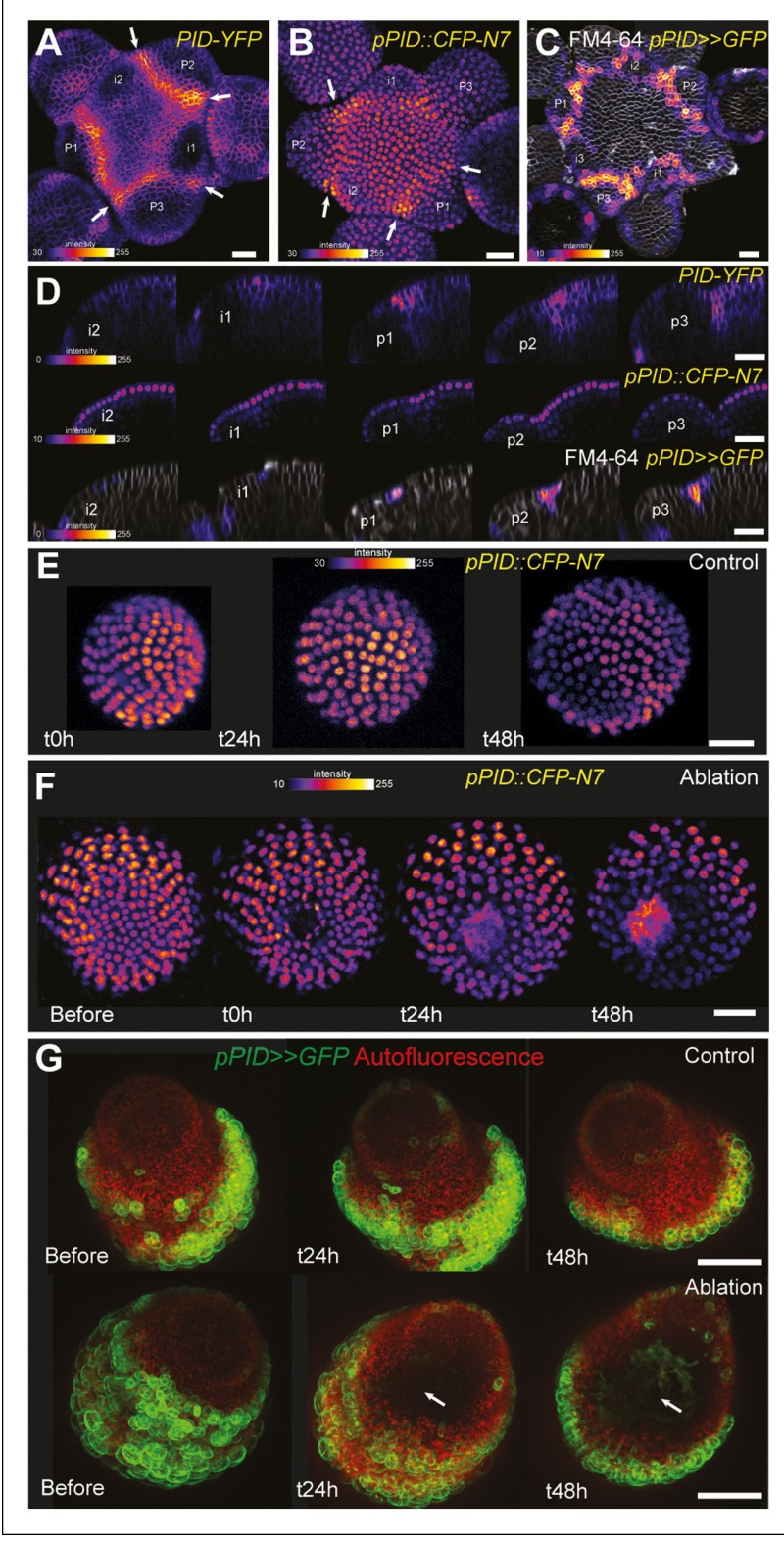

**Figure 6.** *PINOID* promoter activity is not affected by mechanical perturbations. (**A–C**) PINOID expression pattern in representative meristems: a higher expression of PINOID is observed in boundaries. (**A**) Expression pattern of the translational fusion *pPID::PID-YFP*. (**B**) Expression pattern of the transcriptional reporters *pPID::CFP-N7*. (**C**) Expression pattern of *pPID>>GFP*. (**D**) Orthogonal sections through the middle of the boundaries of the meristems presented in (**A–C**). (**E–F**) Time lapse of a representative meristems showing the absence of response of

*Figure 6. continued on next page*

*Figure 6. Continued*

*pPID::CFP-N7* (**F**, n = 13) after ablation when compared to the control (**E**, n = 6). (**G**) Time lapse of a representative meristems showing the absence of response of *pBOUND>>GFP* after ablations (control n = 14, ablation n = 7). Scale bars, 20 μm.
The following figure supplements are available for Figure 6:

**Figure supplement 1.** *pPID* is not significantly induced by isoxaben treatment.
**Figure supplement 2.** *pCUC1* is not significantly perturbed after an ablation in the SAM.
**Figure supplement 3.** *pCUC3* is induced after an ablation in the SAM.

mechanical stress in *STM* expression at the boundary is mediated by local auxin gradients. In other words, whether mechanical stress impacts *STM* expression independently of the response of PIN1 to mechanical stress is unknown.

First, we checked whether mechanical stress can affect the distribution of auxin in the SAM. To do so, we performed a series of ablations on meristems expressing the DII-Venus sensor. Such treatments induce a local reorientation of PIN1 around the ablation (*Heisler et al., 2010*), i.e. parallel to the new stress pattern, and should lead to the local depletion of auxin. After ablation and despite some variability in the basal level of DII-Venus signal on meristems recovering from NPA treatment (*Figure 7A*), we indeed consistently observed an induction of DII-Venus signal around the site of ablation (n = 21; *Figure 7B*, *Video 5*). Interestingly, the first effect of mechanical stress on DII-Venus signal was detected around 4 to 8 h after ablation, which is consistent with the timing of the response of PIN1 after ablation (ca. 4 hr [*Heisler et al., 2010*]) and the subsequent response of *pSTM::CFP-N7* after mechanical perturbation (8 to 18 hr after compression or ablation respectively, this study). The DII-Venus signal was also induced after compression (*Figure 7C*) further matching the response observed in the *pSTM::CFP-N7* line. Altogether, this suggests that auxin patterns and mechanical stress fields act synergistically to control *STM* expression at the boundary.

Next, we investigated whether the mechanical induction of *STM* depends on PIN1. First, we analyzed the DII-Venus signal in the *pin1-6* background. Although polar auxin transport in the meristem is largely inhibited, this mutant can generate bulges and sometimes ill-shaped flowers suggesting that compensatory mechanisms can be induced (*Figure 7D and E*; [*Okada et al., 1991*]). Because some degree of organogenesis is present in the *pin1* mutant, this allowed us to observe boundary formation in this background. To distinguish this form of organogenesis from what we observed in the WT, we use the word 'pseudo-boundary' in *pin1-6*. Despite the variability in DII-Venus signal in the *pin1-6* meristems, we could not detect any DII-Venus signal in *pin1-6* pseudo-boundaries, suggesting that *PIN1* is necessary to deplete auxin from the boundary (*Figure 7D*, n = 6). Interestingly, both *pBOUND>>GFP* and *pSTM::CFP-N7* signals were present in the *pin1-6* pseudo-boundaries (*Figure 7D and E*), thus suggesting that PIN1 is not a major player in the correlation between tissue folding and *STM* expression.

To further confirm this result, we performed mechanical perturbations in the *pin1-6 DII-Venus pSTM:CFP-N7* background. No significant induction of DII-Venus signal could be detected after ablation in *pin1-6* demonstrating that auxin depletion after ablation mainly relies on PIN1 (n = 11, *Figure 7G*; *Figure 7—figure supplement 1*). However, both *pSTM::CFP-N7* and *pBOUND>>GFP* were systematically induced around the site of ablation (*Figure 7F and G*; *Figure 7—figure supplement 1 and 2*). This demonstrates that the induction of *STM* after ablation can occur independently of PIN1. While we cannot rule out the possibility that other auxin carriers substitute for *PIN1* in the *pin1* background, despite our efforts, the induction of *STM* around the ablation site could not be clearly correlated with DII-Venus signal intensity, suggesting that the induction of *STM* after an ablation is not primarily caused by a local reduction in auxin levels. In other words, *STM* expression and auxin distribution can be partially uncoupled and the correlation between tissue folding and *STM* expression always remains.

Last, we used a recently developed protocol in which tissue folding is triggered by the local application of the microtubule depolymerizing drug oryzalin, in the presence of the auxin transport

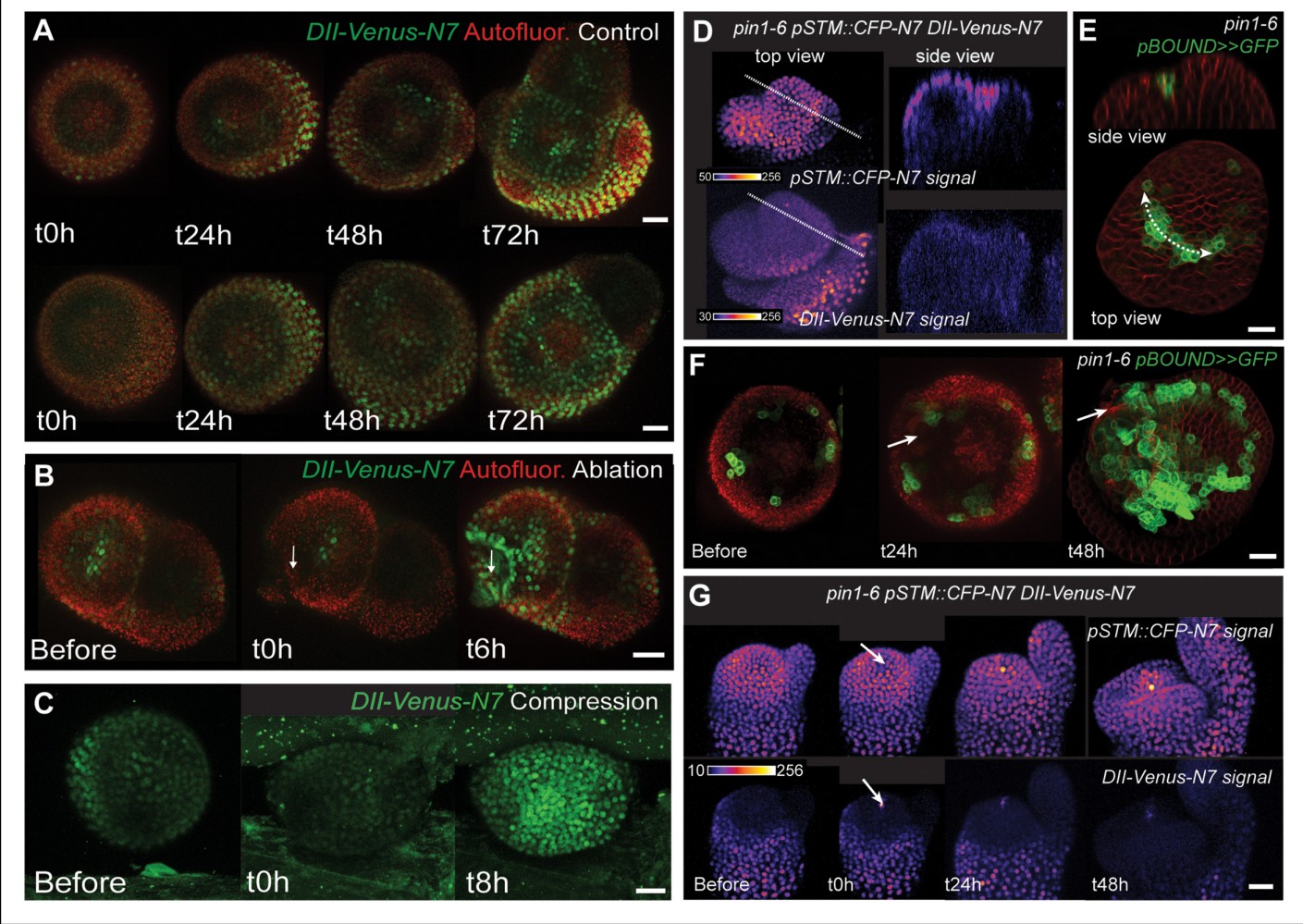

**Figure 7.** *STM* response to mechanical perturbations can be uncoupled from PIN1-dependent auxin distribution. (**A**) Time lapse of representative meristems from a NPA-grown plant and expressing DII-Venus. From t = h, the plants are not exposed to NPA anymore. (**B**) *DII-Venus-N7* signal increases after ablation: Time lapse of a representative meristems from a NPA-grown plant and expressing DII-Venus as in C, after ablation (Control n = 15, Ablation n = 21). (**C**) Representative DII-Venus signal before and after compression. An increased signal is usually detected after 4 to 8 hr after compression in the overall meristem (n = 10). (**D**) Representative *pin1-6* meristem expressing *DII-Venus-N7* and *pSTM::CFP-N7*: the presence of CFP signal at the pseudo-boundary does not correlate with DII-Venus-N7 signal anymore (n = 6). (**E**) Representative *pin1-6* meristem expressing *pBOUND>>GFP* showing the presence of GFP signal in a pseudo-boundary (n = 7). (**F**) Representative *pin1-6 pBOUND>>GFP* meristem after ablation: *pBOUND>>GFP* is induced around the site of ablation (n = 14). (**G**) Representative *pin1-6 DII-Venus-N7 pSTM::CFP-N7* meristem after ablation: *pSTM::CFP-N7* is induced around the site of ablation but *DII-Venus-N7* is not (n = 10). Scale bars, 20 μm.

The following figure supplements are available for Figure 7:

**Figure supplement 1.** *pin1-6 DII-Venus-N7 pSTM::CFP-N7* meristem after an ablation in the SAM.

**Figure supplement 2.** *pin1-6 pBOUND>>GFP* meristem after an ablation in the SAM.

**Figure supplement 3.** *STM* promoter activity during oryzalin-induced tissue folding in the presence of NPA.

inhibitor NPA (*Sassi et al., 2014*). In these conditions, a lateral outgrowth can be induced at the periphery of the meristem, mimicking the first stages of organ emergence. In both the *pBOUND>>GFP* and *pSTM::CFP-N7* lines, oryzalin-triggered tissue folding led to a slight increase of GFP and CFP signals respectively, in the young boundaries (*Figure 7—figure supplement 3*). Because this induction occurred in the presence of NPA and as oryzalin indirectly modifies the

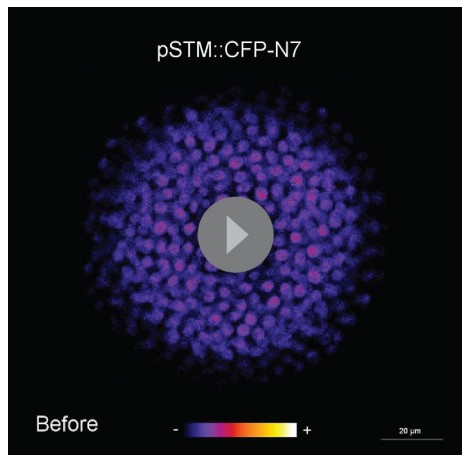

pSTM::CFP-N7

Before

20 μm

**Video 5.** DII-Venus-N7 after an ablation (relates to *Figure 7B*).

mechanical status of cell walls (and thus promotes organ emergence), this further supports a scenario in which mechanical stress can promote the expression of *STM* at the boundary, relatively independently of auxin distribution.

## Discussion

Because tissue folding is generally associated with the presence of high mechanical stress levels, crease formation is a unique event in development where the biochemical regulation of morphogenesis may also involve a strong contribution of mechanical signals. For instance, the patterning genes *Twist* and *Notail* are induced by the deformation of the embryo during gastrulation and epiboly in Drosophila and Zebrafish embryos respectively, through the activation of the β-catenin pathway (*Brunet et al., 2013*; *Farge, 2003*; *Desprat et al., 2008*). This suggests that shape changes are not only the result of biochemical regulation, but are also the source of mechanical signals that further channel morphogenesis *via* the control of gene expression. Our observations are in line with these conclusions: we report that the expression of *STM* at the boundary of the shoot apical meristem is correlated with tissue folding and is increased by mechanical stress. This provides a scenario in which biochemical factors, such as auxin, promote differential growth and shape changes in the meristem, which in turn, generate mechanical signals that can impact the expression of some of these regulators.

Incidentally, our work also echoes results from Drosophila embryo where tension lines acting as genetic and mechanical boundaries have been revealed in the wing disk and shown to compartment cell identities (*Aliee et al., 2012*; *Landsberg et al., 2009*). Here we show that mechanical stress at the boundary contributes to the local expression of *STM* that is in turn necessary for the separation between organs. Note that tensile stress at the meristem boundary becomes anisotropic before tissue folds, as maximal stress direction is first prescribed by differential growth. In this respect, our results might thus be extended to other tissues like leaves or embryo, where the induction of *KNOX* genes would be related to differential growth, and thus stress, before tissue folding.

Beyond organ separation, the boundary has many other functions. In particular, it is the site where new meristems, called axillary meristems, can be initiated. Interestingly, overexpression of class I *KNOX* genes is sufficient to induce ectopic meristems (*Chuck et al., 1996*). The boundary also plays a crucial role in plant architecture: this is where auxin is redistributed from the adjacent neighboring organ to the meristematic pool where new organs or new axis need to be initiated (*Heisler et al., 2005*; *Reinhardt et al., 2003*; *de Reuille et al., 2006*). The central role of the meristem boundary in plants largely explains why this domain has received considerable attention over the past decades, albeit from a molecular genetics point of view mainly (*Aida and Tasaka, 2006*; *Tian et al., 2014*). The role of mechanical stress in the function and regulation of the boundary domain of the meristem opens the possibility that other functions than organ separation are influenced by mechanical signals.

Another question raised in this work is that of the coordination between mechanical and biochemical signals to control development. In animals, well-known elements of biochemical-based transduction pathways are involved in mechanotransduction. For instance, integrin, β-catenin or YAP/TAZ are all involved in transducing mechanical signals (e.g. *Janmey et al., 2013*; *Jaalouk and Lammerding, 2009*; *Orr et al., 2006*). Such intermingling raises the question of whether these signals can really be uncoupled with one another and this may even question the added value of mechanical signals in development. This is what we touched upon by analyzing the coordination between the plant hormone auxin and mechanical stress in the meristem. While we provide further evidence supporting a strong coordination between *STM* expression and auxin depletion, we also found that the response to mechanical stress can uncouple them. In particular, mechanical

perturbations induce *STM* expression even when auxin transport or level is affected. In addition, mechanical stress does not seem to impact the expression of *PINOID*, which has been associated with the control of PIN1 polarity and thus auxin patterns in the meristem (*Robert and Offringa, 2008*). This is thus a case where two redundant signalling pathways act relatively independently to control the same morphogenetic event. We propose that the absence of a strict coupling between auxin depletion and mechanical stress is a way to add robustness in meristem functions at the boundary. Incidentally, this highlights the added value of mechanical signals in this domain.

Alternative cues may be involved in the promotion of *STM* expression at the boundary. In addition to auxin, *STM* has been associated with the homeostasis of cytokinins, gibberellins and ethylene. Activation of *STM* promotes the expression of the cytokinin biosynthesis gene *AtIPT7* and cytokinin response factor *ARR5* (*Yanai et al., 2005*). Conversely, overproduction or application of cytokinins increase the expression level of *STM* and can rescue weak *stm* mutant alleles (*Rupp et al., 1999*; *Jasinski et al., 2005*). This interaction with cytokinins is largely shared among the different class I *KNOX* genes and, consistently, it has been shown that *KNAT1* and *KNAT6* display redundant functions with *STM*. Interestingly, the level of cytokinins has been predicted to increase in the boundary, based on the cytokinin activity reporter pTCS signal, and axillary meristem formation has been shown to require cytokinin signalling (*Wang et al., 2014*), consistent with the well-known bushy phenotypes of cytokinin overproducers. Temporally, this cytokinin burst would follow the initial reduction in auxin level at the boundary (*Wang et al., 2014*). Thus cytokinin, in parallel to and after auxin depletion, may very well contribute to *STM* induction at the boundary. Class I KNOX proteins also reduce gibberellins levels, notably through a direct repression of the GA-20 oxidase gene (*Sakamoto et al., 2001*; *Chen et al., 2004*); conversely gibberellins can rescue the phenotype of KNOX overexpressors (*Hay et al., 2002*). The induction of the gibberellin catabolism gene GA-2 oxidase was also shown to depend on KNOX proteins (*Jasinski et al., 2005*).

While both cytokinins and gibberellins are tightly linked to class I KNOX gene expression, the potential interplay with mechanical stress has not been explored. In contrast, other hormones, such as ethylene, have been associated to the plant response to many stresses. Interestingly, ethylene restricts the expression of class I KNOX gene *KNAT2* in the shoot meristem to the boundary domain (*Hamant et al., 2002*). Note however that the exact contribution of ethylene in mechanoperception remains debated. For instance, while touch can induce ethylene synthesis, ethylene response mutants still display thigmomorphogenesis in response to touch (*Chehab et al., 2012*); touch-induced thigmomorphogenesis rather depends on jasmonate (*Chehab et al., 2012*). While certain hormones are rather associated with repetitive elastic deformations (like touch or wind), others are involved in continuous plastic deformations (i.e. growth and shape changes), these may be better candidates for the regulation of *STM* expression at the boundary.

More generally, and as auxin does not seem to act as a secondary messenger of mechanical signals for *STM* expression, another exciting prospect for this work will be to identify the elements of the mechanotransduction pathway acting on *STM* at the boundary. The current dissection of the gene regulatory network acting at the boundary (e.g. [*Tian et al., 2014*]) should help us weigh the putative contributions of these biochemical factors in transducing mechanical signals at the SAM in the future.

As illustrated in the art of origami, folding can in principle generate the widest diversity of shapes from simple inputs. In the past decades, there has been tremendous progress in the identification of genes that trigger such events in all living organisms. The robustness of such a complex regulation is often thought to rely on redundancy between different molecular pathways. Here we have investigated whether the shape itself, and its associated mechanics, interferes with gene expression and channels morphogenesis by constraining the possible outputs of the gene regulatory network. If generalized, such a dialog between gene and form could help us understand how reproducible shapes can emerge from a complex gene regulatory network that is susceptible to noise. Because of its essential function in morphogenesis and its impact on developmental robustness, this 'shape to gene' feedback may also have far reaching implications in evolution.

## Material and methods

### Plant lines and growth conditions

The *pBOUND>>GFP, DII-Venus, pin1-6, stm-dgh6* lines were already described in the literature (*Brunoud et al., 2012*; *Vernoux et al., 2011*; *Laufs et al., 2004*; *Dharmasiri et al., 2005*; *Vernoux et al., 2000*; *Aida et al., 2002*). The *pCUC1::CUC1-GFP* and *pCUC3::CFP* lines have recently been described (*Goncalves et al., 2015*). Note that the *Mo223::GFP* line (*Cary et al., 2002*) was also used to test the response of the *CUC1* promoter to ablations. The *pJAZ10::GUS* used as control for the JA treatment has also been previously described (*Mousavi et al., 2013*). The *pSTM:: CFP-N7, pPDF1::CFP-N7* and *pPID::CFP-N7* lines were generated by fusing either the 5,7 kb region upstream of *STM* (AT1G62360) ATG, the 1456 pb region upstream of *PDF1* (AT2G42840) ATG, or the 4 kb region upstream of *PID* (AT2G34650) ATG to a *CFP* targeted to the nucleus through a *N7* signal in a *pH7m34GW* vector using the Gateway system and transformed in *Col0* ecotype. The *pPID>>GFP* line was generated by fusing the 4 kb region upstream of *PID* (AT2G34650) ATG to the Alc-R coding sequence fused to an *pAlcA::GFP-ER* in a derivate of the *pGREEN129* plasmid and transformed into the WS-4 ecotype. The *pSTM::STM-Venus* line was generated by transforming the plasmid described by Heisler *et al.* (*Heisler et al., 2005*) in *Col-0* plants. The *STMamiRNA* line (binding to the *STM* coding sequence though the following sequence: TTAACCACTGTACTTGCGCGA) was designed and amplified from the *pRS300* plasmid following previously described protocol (http://wmd3.weigelworld.org/cgi-bin/webapp.cgi) and fused to the *pSTM* promoter in a *pK7m34GW* vector using the Gateway system. The construction was transformed either in Col-0or in the *pSTM::STM-Venus* line. *pBOUND>>GFP and pPID>>GFP* expression was induced by an overnight treatment with 70% ethanol vapor following a previously described protocol (*Deveaux et al., 2003*).

'Greenhouse grown plants' were initially grown in short-day conditions (8 hr/16 hr light/dark period) for one month and then transferred to long-day conditions (16 h/8 h light/dark period). Stems were cut and the SAM was dissected when the inflorescence meristem was visible, i.e. between the appearance of the first flower to the appearance of first silique (stages 13 to 17 [*Smyth, 1990*]) and transferred on a half MS medium with vitamins and 0.125 µg/µL of BAP for imaging as already described (*Fernandez et al., 2010*).

"In vitro grown plants" were grown in a phytotron in long day conditions on *Arabidopsis* medium (Duchefa, Haarlem, the Netherlands) supplemented with 10 µM NPA to inhibit flower initiation and generate naked meristems. NPA-treated in vitro grown plants were transferred to a medium without NPA as soon as naked meristems were formed as already described (*Grandjean et al., 2004*). Meristems were then imaged from 24 hr to 48 hr after transfer on the NPA-free medium. For in vitro experiments on *pin1* meristems, plants were grown in vitro on MS medium without NPA in a phytotron in long day conditions.

### Confocal laser scanning microscopy and image analysis

Dissected meristems and plants grown in vitro were imaged in water using either a LSM700or a LSM780 confocal microscope (Zeiss, Germany) to generate stack of optical sections with an interval of 0.25, 1or 2 µm between slices. In some cases, membranes were stained with FM4-64 and the signal in the L1 was extracted using the Merryproj software as described in de Reuille *et al.*, (*de Reuille et al., 2005*). Longitudinal optical sections of various thicknesses were performed using the ImageJ software. In many cases, the Fire lookup table from ImageJ was used to represent the *CFP* signal intensity and background signal from raw images was reduced in images, by modulating the minimum level of intensity in ImageJ as shown by the calibration bars on the different pictures. The minimum level was however never above 50 out of 256 in intensity and the same processing was applied on all time points, and on control and assay alike. In some projections from of *pSTM::STM-Venus* and *pSTM::CFP-N7* where the signal was weak, area outside the nucleus emitting in both green and red channels and corresponding to either the auto-fluorescence of the plasts or the FM4-64 signal were automatically removed using the function calculator-plus subtraction of ImageJ.

The maps and quantifications of meristem curvature and *pSTM::CFP-N7* signal at cellular levels were obtained using the MorphographX software (www.morphographix.org). The curvature maps were generated by plotting the mean Gaussian curvature on either non-segmented or cell-

segmented meshes with a neighboring of 15 μm. The maps of *pSTM::CFP-N7* signal were generated by blurring the CFP signal with a radius of 10 μm and projecting it on the cellular mesh with a thickness of 10 μm. Note that the oldest organs were not included in the quantifications because of the inability of the mesh to follow the surface of the meristem in very deep boundaries and to extract well the *pSTM::CFP-N7* signal in these area. Furthermore, cells at the extremities of the mesh were also not included in the quantifications due to the presence of aberrations of Gaussian curvature on these parts of the mesh.

Organ size was quantified using ImageJ: stacks were projected using the function z-stack, a circle was drawn using the freehand selection around each organ viewed from the top and the surface inside this selection was automatically measured. The folding angle of the boundary was measured manually on longitudinal sections taken in the middle of each boundary using the angle tool of ImageJ.

For the quantification of the *pSTM::CFP-N7* nuclear signal after Isoxaben treatments or after ablations, each nucleus from the central zone (Isoxaben treatment) or from the peripheral zone (around the site or on the opposite site of the ablation) was isolated manually using the ImageJ software on the slice from the z-stack where its size and intensity was the highest. A circle was manually fitted around each nucleus and the integrated intensity was calculated in this circle. 10 nuclei were measured for each meristem. The quantification of nucleus signal intensity after Isoxaben treatment was performed on one of two sets of individual plants (see *Supplementary file 1*).

## Compressions, ablations and chemical treatments

Each experiment was performed on at least two independent sets of plants, and with at least 4 independent plants in each set. In all experiments, the t = X hr time point corresponds to X hours after the beginning of treatment. Controls and assays were analyzed in parallel (same growth conditions, same imaging conditions). The compressions, ablations and isoxaben treatments that were carried out on WT plants were performed on plants previously grown in vitro NPA and transferred in a medium without NPA 0 to 24 hr before the beginning of the experiment.

The ablations were performed with a needle or using a pulsed UV laser as already described (*Hamant et al., 2008*; *Uyttewaal et al., 2012*). Note that the ablations performed on *pin1-6* meristems were performed on plants grown in vitro on NPA-free medium.

The isoxaben treatments were conducted by immersing the plants in solutions of 5 to 20 μM of isoxaben overnight (for 12 to 14 hr, *Heisler et al., 2010*; *Uyttewaal et al., 2012*). Controls were obtained by water immersion with an equivalent volume of Dimethyl Sulfoxide (DMSO). All these concentrations gave similar results. The presence of isoxaben in the meristem could be confirmed by its impact on meristem, cell and nucleus size.

For the auxin treatments, dissected meristems in half MS Vitamins and BAP boxes were immerged in a solution of 10 μM of 2-4D overnight (for 12 to 14 hr). Controls were obtained by immersing the meristem in a solution of equivalent volume of water. Similar experiments were also performed on plants grown in vitroand recovering from an NPA treatment and with IAA and gave similar result (data not shown). The disappearance of the *DII-Venus-N7* signal reflected the presence of high auxin levels in treated meristems.

For jasmonate treatments on *pSTM::CFP-N7* and *pJAZ10::GUS*, jasmonic acid (Sigma) was diluted in 100% Ethanol to make a stock solution. JA was then diluted in water (1/1000) for the treatments. NPA grown plants in *Arabidopsis* medium exhibiting naked meristems and recovering from the NPA treatment were fully immersed overnight (12 to 14 hr) in a water solution containing 100 μM of JA or in a similar dilution of ethanol for the control.

Oryzalin applications in lanolin paste were carried out as previously described (Sassi et al., 2014).

## GUS staining

To reveal the presence of GUS activity in the *pJAZ::GUS* plants grown on NPA and treated with jasmonate, the samples were first immersed overnight at 37°C in a GUS solution (100 mM $Na_2HPO_4$ pH7, 5 mM $K_4Fe [CN]_6 \cdot 3H_2O$, 5 mM $K_3Fe [CN]_6$, 0.05% Triton-X-100, 0.5 mg/ml X-Gluc) before being cleared trough successive ethanol baths (from 70 to 100% Ethanol). Imaging was performed with a Zeiss stereomicroscope Discovery V20.

## Whole mount immunolocalizations of PIN1

The whole mount immunolocalizations of PIN1 were performed on WT plants (*WS-4* ecotype) as previously described (*Besnard et al., 2013*). Imaging was performed with a Zeiss LSM700 microscope.

## Whole mount *in situ* hybridization

The whole mount in situ hybridization of *STM* mRNA was performed as previously described (*Rozier et al., 2014*). Imaging was performed with a Zeiss stereomicroscope Discovery V20.

### *Stm-dgh6* genotyping

The *stm-dgh6* mutant was originally identified in the Versailles T-DNA collection. In this allele the T-DNA is inserted between nucleotide 723 and 724 in the cDNA sequence. To genotype the *stm-dgh6* populations, we used the forward primer GGGTAAATACCCCTTTGATGG (before the T-DNA insertion) and reverse primer TCGTTCCTTATCCTGAGTTG (in the following gene). This amplified a 2600 bp long fragment in the WT and *stm-dgh6* heterozygote, and nothing in the *stm-dgh6* homozygote. Conversely, to detect the presence of the T-DNA, we used the forward primer CGTGTGCCAGGTG-CCCACGGAATAGT (LB4, T-DNA) and reverse primer GTAGTGACGGCTCCACCAAT (*STM*). This amplified a 300 bp long fragment in the *stm-dgh6* heterozygote and homozygote, and nothing in the WT.

## qPCR on STMamiRNA lines

The qPCRs for *STM* and *STMamiRNA* were performed on 2 week-old seedlings grown on ½ MS medium with kanamycin. The qPCRs on the *pSTM::STMamiRNA* lines were performed on either homozygous or heterozygous F3 populations. The *qPCR* on WT plants was performed on plants harboring a kanamycin resistance at homozygous state and growing in the same conditions as the *pSTM::STMamiRNA* lines. As shown in *Figure 3—figure supplement 2*, the expression of the amiRNA (Primers: *STMamiRNA Q5*: CACTGGTTATTCACAGGTCGTG, *STMamiRNA Q3*: CAGTGGT-TAATCAAAGAGAATCAATG) and its effect on *STM* expression was checked by qPCR (Primers: *qSTM-F*: TCGACTTCTTCCTCGGATGACCCA, *qSTM-R*: TCTCCGGTTATGGAGAGACAGCAA). Three independent biological replicates were at least used for each sample. Expression is shown relative to that of *TCTP*, a stable housekeeping gene.

## Acknowledgments

This work was supported by Agence Nationale de la Recherche ANR-10-BLAN-1516 and ANR-10-BLAN-1614, by the European Research Council ERC grant 615739 'MechanoDevo' and ERC grant 294397 'Morphodynamics'. We thank Henrik Jönsson, James Locke and Elliot Meyerowitz for allowing B.L. to carry some experiments while working on a new project at the Sainsbury laboratory. We thank Frédéric Bègue for help with the initial compression experiments on *pBOUND>>GFP* and Isabelle Bohn-Courseau for contributing to the generation of the *pPID>>GFP* line; Véronique Boltz for help with molecular biology; Weibing Yang for help with in situ hybridization; Fabrice Besnard, Marina Oliva, Géraldine Brunoud and Magalie Uyttewaal for providing unpublished material and for advice. We also thank Platim (UMS 3444 Biosciences Gerland-Lyon Sud) for help with imaging, the Sainsbury lab in Cambridge for access to a pulsed UV laser system and Centre Blaise Pascal (CBP, ENS de Lyon) for help with computational analysis.

## Additional information

### Funding

| Funder | Grant reference number | Author |
| --- | --- | --- |
| Agence Nationale de la Recherche | ANR-10-BLAN-1516 | Benoît Landrein<br>Annamaria Kiss<br>Jan Traas<br>Arezki Boudaoud<br>Olivier Hamant |
| European Research Council | ERC-2013-CoG-615739 | Olivier Hamant |

| Agence Nationale de la Recherche | ANR-10-BLAN-1614 | Millan Cortizo Patrick Laufs |
| --- | --- | --- |
| European Research Council | ERC-2013-AdG-294397 | Massimiliano Sassi Jan Traas |

The funders had no role in study design, data collection and interpretation, or the decision to submit the work for publication.

## Author contributions

BL, Major conceptual and experimental contribution, Wrote the manuscript, Conception and design, Acquisition of data, Analysis and interpretation of data, Drafting or revising the article, Contributed unpublished essential data or reagents; AK, Analysis and interpretation of data, Contributed unpublished essential data or reagents; MS, Contributed the analysis of STM expression on oryzalin + NPA, Acquisition of data, Analysis and interpretation of data; AC, Acquisition of data, Analysis and interpretation of data; PD, Contributed to the design of the LSM method; MC, ST, MA, Contributed unpublished essential data or reagents; PL, Provided the unpublished pPID>>GFP line; JT, Discussed the work with other co-authors, Supervised M. Sassi, Inputs on the draft; TV, Discussed the work with other co-authors, Contributed unpublished material, Inputs on the draft; AB, Discussed the work with other co-authors, Inputs on the draft; OH, Initial experiments and conceptual design, Wrote the manuscript, Conception and design, Acquisition of data, Analysis and interpretation of data, Drafting or revising the article

## Author ORCIDs

Massimiliano Sassi, http://orcid.org/0000-0002-9685-4902

# Additional files

**Supplementary files**

• Supplementary file 1. Summary of the number of replicates for each mechanical test.

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
