## [Decision Letter]

Thank you for sending your work entitled "Mechanical stress channels *STM* expression in the *Arabidopsis* meristem" for consideration at *eLife*. Your article has been evaluated by Detlef Weigel (Senior Editor), a member of our Board of Reviewing Editors, and three reviewers, one of whom, Gabriel Monshausen (Reviewer 3), has agreed to share his identity.

We are in principle interested in the work, but the reviewers have also identified a series of concerns. Below you can find the individual comments of the three reviewers, and we would like to ask you to address them point-by-point.

In the discussions between the reviewers and editors, a few key points emerged. Please pay particular attention to these:

1) A key aspect of your paper is your claim that *STM* expression is mechanosensitive. However, given the considerable lag between the mechanical insult and the change in *STM* expression, the reviewers feel that alternative interpretations are still valid and the claim that the *STM* promoter is mechanosensitive is overstated. We would therefore like to ask you to perform a more comprehensive analysis to strengthen your idea. For example, how does *STM* expression react to transient mechanical stimuli, such as pinching? Moreover, we would like to ask you to perform experiments to exclude the possibility that the observed expression changes are due to a wounding response. Analysis of *STM* expression upon local jasmonate application is one experiment that can be helpful here. Demonstrating that the behavior of the *STM* reporter is unchanged in the JA signaling mutant *coi1* would be another option. (We are aware that others have suggested that JA may be involved in thigmotropism, but it appears that those effects are probably rather indirect).

2) There are some questions about the specificity of your observations. If there is indeed causality linking mechanical stress and *STM* expression rather than correlation only, one could expect that this does not hold for any gene expressed in the boundary region. Performing the key experiments with other genes that are expressed in the boundary region could answer this question. Such control experiments should focus on other transcriptions factors.

3) Please also carefully address some technical issues. It would be especially important to demonstrate that your assumption that the *amiRNA* construct preferentially downregulates *STM* in the boundary region is true. Combination of this construct with a *STM* protein fusion reporter could substantiate this claim.

*Reviewer #1:*The authors of this paper are attempting to advance the hypothesis that mechanical stress in the shoot apical meristem turns on expression of the *STM* transcription factor which then acts to mediate the separation between organs.

To do this, the authors do experiments to:

1) Show a correlation between *STM* reporter expression and curvature.

The authors do a convincing job of establishing this correlation.

2) Show that the *STM* reporter is activated by mechanical stress.

Mechanical stresses are applied to the apical meristem in the form of ablation or compression. *STM* reporter expression increases but with a long lag such that the authors conclude this is a secondary effect of mechanical stimulation. They attempt to connect the mechanical stimulus to alterations in *STM* reporter expression via the auxin pathway but experiments show that the mechanically induced changes in the auxin pathway cannot easily explain the changes in *STM* expression.

3) Show that *STM* expression in the boundary zone is required for organ separation.

The requirement for *STM* function in organ separation has been shown previously (Long and Barton, Development, 1998; Belles-Boix et al., 2006) making this experiment largely confirmatory.

While the authors might argue they specifically knock down *STM* function in the peripheral zone, this is complicated by the observation by other labs (e.g. Jackson lab, Kim et al., 2003, Development) that *STM* protein can move from its site of synthesis into adjacent cells. Therefore, using a microRNA in the boundary zone would affect the action of *STM* in both this zone as well as in any cells into which *STM* moves.

In summary, curvature and *STM* reporter expression are correlated in the shoot apical meristem. However, the experiments in this paper do not satisfactorily solve the question of causation. Because *KNOX* genes can cause alterations in curvature in otherwise smooth surfaces (e.g. Tsiantis lab work in Cardamine leaf; Long and Barton, 1998, Development, for work in the *Arabidopsis* embryo), the simplest explanation seems to be that *STM* causes the alterations in curvature by inhibiting cell division and/or cell expansion at the boundary. A mechanism for this is suggested by findings by the Hake lab showing that KNOX targets include genes involved in auxin transport as well as genes involved in cell wall synthesis.

As for the experiments showing that mechanical stress induces *STM* expression, it is possible that many types of wounding or perturbations to the shoot apical meristem will cause an indirect increase in *STM* reporter expression as the plant attempts to compensate for the insult. The authors should test additional stresses to see if this is a more general phenomenon. The authors should also extend the analysis to other "meristem genes" expressed especially in the boundary zone – *CUCs, BOPs, JLO* – to determine if this is specific to *STM*.

*Reviewer #2:*The core message of this very interesting manuscript is that mechanical stress activates the expression of *STM*, a homeodomain transcription factor with key roles in maintaining meristem fate at the boundary of meristem and organ primordial and that *STM* is required for boundary formation and organ separation. The finding that an important developmental regulator is under control of physical forces in the growing organism while at the same time acting on shaping the organism is certainly of relevance to a wide audience. The experiments (mostly live cell imaging of fluorescent reporters, coupled to perturbations) and analyses shown are of high quality and are mostly well presented. However, in the present form the manuscript has several important shortcomings, which need to be addressed, but based on the exciting findings, I think it is a strong candidate for publication in *eLife* after revision.

1) The authors exclusively use a fluorescent reporter as proxy for *STM* activity, however, it does not faithfully recapitulate endogenous *STM* expression. The differences need to be addressed, preferably by in situ hybridization of reporter plants using *STM* and CFP probes. I think this experiment is essential validate the major claims concerning the responsiveness of *STM* expression to mechanical stimuli. Along these lines, I would also like to see that endogenous *STM* expression responds to a selected experimental stimulus, such as pinching. Again, these experiments could be done in the reporter line, comparing *STM* and CFP expression by qRT-PCR. These experiments would unequivocally demonstrate that the observed CFP induction is not an artifact of the reporter.

2) The boundary phenotype in the *pSTM::STM* rescue and *amiRNA* silencing plants is not documented properly. To be able to judge the phenotypes, we need SAM microscopy and curvature measurements. The pictures shown in Figure 3 are not very informative with regards to the boundary effect, rather they show overall growth habitus.

3) The boundary specific silencing supposed to be happening in the *amiRNA* lines is not documented. Again, in situ hybridization seems essential to analyze where *STM* silencing occurs, to be able to interpret the results shown in Figure 4.

4) The claim the *STM* responds to mechanical stimuli after 8 hours onwards is not substantiated by the data presented. From the images shown in Figure 5 it seems that the first effects become visible only after around 20 hours. Please comment.

5) The authors routinely refer to "*STM* expression" when they are talking about *pSTM::CFP* reporter activity. Please be more careful with wording.

6) The claim that *STM* reporter expression is much higher after removal of NPA is not substantiated by the data shown in Figure 1—figure supplement 1.

*Reviewer #3:*This interesting manuscript by Hamant and coworkers addresses the question of how the expression of *STM*, a key regulator of the shoot apical meristem also required for organ separation, is regulated. This is an important biological problem, which is explored by the authors using a range of sophisticated imaging, image processing and genetic tools. Overall this manuscript is very well written and the data provide strong experimental support for a role of mechanical stress in controlling *STM* expression patterns.

1) To analyze the relationship between mechanical stress, and more specifically Gaussian curvature, and *STM* expression, the authors mapped meristem curvature at high resolution and projected this onto a map of *pSTM::CFP-N7* expression. These experiments convincingly show that negative Gaussian curvature is correlated with enhanced *STM* expression in the boundary domain of the SAM.

2) The authors further analyze whether *STM* expression in the boundary is functionally important for organ separation during development. To this end, *STM-Venus* preferentially expressed in the boundary was introduced into a strong *STM* mutant (*dgh6*). The resulting transgenic lines exhibited only a partial recovery of meristem maintenance but a complete loss of organ fusion defects. *pSTM::STMamiRNA2* was introduced into WT to preferentially silence *STM* expression in the meristem boundary. This resulted in reduced, but functional meristems, and major organ fusions.

These are creative attempts to specifically alter *STM* expression in the boundary region; however, no evidence is presented to actually confirm that such preferential alteration has actually occurred. It would have been informative to show the expression pattern of *pSTM::STM-Venus* not only in a WT background (Figure 3) but in the *dgh6* mutant line, as the expression pattern of the weakly expressing WT parent line (Heisler et al. 2005a) may not necessarily have been maintained after crossing. Similarly, it would be useful to know whether *STM* expression is preferentially silenced in the boundary in *pSTM::STMamiRNA2* lines. Transformation of the constructs into *pSTM::STM-Venus* lines would have been informative. If these experiments are not done, the limitations should at least be discussed.

3) The authors convincingly demonstrate that enhanced *STM* expression in the boundary is correlated with but not dependent on local auxin depletion. Treatment with 2,4-D globally increased auxin levels in the SAM (as monitored using DII-Venus) but did not alter the pattern of *STM* expression (although slightly increasing the intensity of *pSTM::CFP-N7* expression).

One question this experiment raised for me was whether 24 hour incubation with 2,4-D altered meristem organization in any noticeable manner, i.e. were Gaussian curvatures in the boundaries affected by the treatment? According to the authors' model, this should have affected *STM* expression.

4) The authors provide strong evidence that mechanical perturbations (compression of meristem and local cell ablation) and pharmacological treatments resulting in altered mechanical stress (isoxaben, NPA+oryzalin) resulted in increased *STM* expression. Based on the authors' initial experiments, *STM* expression was correlated with negative Gaussian curvature. Does this relationship hold when the SAM is perturbed by the various treatments described here? This analysis could greatly strengthen the concept that *STM* expression is regulated by a specific type of mechanical stress.

[Editors' note: further revisions were requested prior to acceptance, as described below.]

Thank you for resubmitting your work entitled "Mechanical stress channels *STM* expression in the *Arabidopsis* meristem" for further consideration at *eLife*. Your revised article has been favorably evaluated by Detlef Weigel (Senior Editor), a Reviewing Editor, and three reviewers, one of whom, Gabriel Monshausen (Reviewer 3), has agreed to share his identity. The manuscript has been improved but there are some remaining issues that need to be addressed before acceptance.

All reviewers and the editors agree that your work is interesting and original, but as you can see from their comments, the reviewers still have some questions about the causality relations put forward in your study. Yet, there is also consensus that this is a hard chicken-and-egg type of problem, and a completely unambiguous answer is hard to come by with current means. Although things might turn out somewhat different in the long run, your study presents one of the best ways forward into this area for the moment, and is likely to stimulate further research. In light of this, we believe that it would be in your own interest to be very careful how you phrase your conclusions. Please go once more through your manuscript and tone down any strong statements with respect to situations that entail a possibly circular logic. In that respect, please pay particular attention to the comments of reviewer 1.

Please also clarify the following: there is some doubts whether the experiments show jasmonate insensitivity of *STM* expression as stated in the text. In the supplemental figure, there is clear *STM* expression increase that does not occur in any of the mock treated samples. It thus appears simply wrong to say it is insensitive, and these results should not be "hidden" in the supporting information.

*Reviewer #1:* This paper presents several interesting observations. My problem with the paper – even after the changes and in some cases even more so after the authors' additions – is that the authors shoehorn their observations into a model in which mechanical stress increases *STM* expression.

In the body of the paper, the authors include alternative explanations for the correlation such as in this paragraph:

"The fact that *STM* expression correlates with curvature at the boundary can simply be explained by the fact that *STM* reduces the growth rate at the boundary, leading to tissue folding. *KNOX* genes can cause alterations in curvature in otherwise smooth surface (Long and Barton, 1998, Barkoulas et al., 2008). The impact of *STM* on crease formation at the boundary could be mediated by inhibiting cell growth in this domain. Consistent with this scenario, KNOX target genes include genes involved in auxin transport as well as genes involved in cell wall synthesis (Bolduc et al., 2012). However the strong correlation between *STM* promoter activity and tissue folding suggests that a signal, which is related to curvature, could add robustness to the *STM* expression pattern at the boundary."

Why would a strong correlation lead one to prefer one direction of causality over the other? We know that *STM/KNOX* expression exists in the absence of tissue folding and also precedes tissue folding in the embryo, and precedes tissue folding in the development of leaf lobing. To my knowledge, there is no case of tissue folding that precedes *STM/KNOX* expression. Of all the treatments that the authors subject the plants to, pinching would seem to best simulate folding but pinching the meristem does not increase *STM* expression. This argues strongly for the arrow of causation to be in the direction counter to what the authors propose. (The authors make arguments about how pinching is somehow different than the stresses that would normally be sensed by *STM* but I am not convinced.)

Whatever the reality may be, in this paper alternative interpretations are downplayed. There is no mention of alternative models in the Discussion and the title is overly strong in its assertion given the weak, and by the authors' own admission, indirect, connection between physical stress and *STM* expression.

Unfortunately, without the framework of mechanical stress acting to increase *STM*, the experiments presented in this paper lack cohesion.

Comments on revisions:

1) The authors shore up their complementation data. They have now used PCR to genotype homozygous *stm/stm; pSTM::STM-Venus/pSTM::STM-Venus* double mutants segregating from among the progeny of *stm/+ pSTM::STM-Venus/pSTM::STM-Venus* parents. It would have been more convincing if they had recovered a fertile double homozygote but if the PCR data are compelling, then they likely have identified complemented individuals. (Unfortunately, the PCR data aren't shown. I would be worried about using the presence/absence of a 2.6 kb band for determining the presence of the wild type allele as lack of a band can be due to simple PCR failure. Presumably the authors have adequately controlled for this.)

2) The authors have added a test for response of *STM* expression to jasmonate, a hormone associated with wounding stress. This is an experiment to determine whether the response of *STM* expression to ablation might be in response to wounding. The data are given in the supplemental section (Figure 5–figure supplement 3). Most plants do not respond – however a significant fraction, 4 out of the 20 or so plants tested, *did* show an increase in *STM* expression in response to jasmonate. The authors dismiss this positive result as unimportant but I am not sure what the basis for their dismissal is.

(I can easily envision an entirely different historical scenario for these experiments that would lead to a different set of conclusions: A set of jasmonate researchers could have reasonably concluded that applied jasmonate can increase *STM* expression. They would then have followed up that observation with ablation experiments on the meristem finding that *STM* expression increased in response to wounding.)

In short, the jasmonate experiments indicate that the situation is likely to be complex. Rather than sweeping this complexity under the rug, the authors should discuss it.

3) The authors add data for other boundary-expressed genes and find that one – *CUC3*, but not the other, *CUC1*, respond to ablation with increased expression. This is very interesting but I am not entirely sure what to make of it. Again, it reinforces the conclusion that the situation is complex.

(I would add here as well that these experiments are done in NPA treated meristems and therefore the results hold for a context in which auxin is improperly localized. Since there is reason to believe that auxin and *KNOX* genes interact at some level, this is important to keep in mind. A different result might be obtained in the context of properly localized auxin).

*Reviewer #2:*I am very happy with the thorough revision of the manuscript and the addition of many meaningful and well-executed new experiments. I now support publication of the manuscript without hesitation.

The only issue I still have is the rather poor quality of *STM* in situ hybridizations after ablation. With some good will it is possible to identify enhanced *STM* RNA levels next to the ablation site, but it could be helpful to include more apices (if the authors have pictures available) to make that trend clearer.

*Reviewer #3:*The authors satisfactorily addressed the issues that I raised previously; the manuscript is considerably strengthened by new experiments (i) showing insensitivity of *STM* expression to JA, (ii) confirming preferential downregulation of *STM* in the boundary region by *STM* amiRNA and (iii) demonstrating differential mechanosensitivity of *CUC1* and *CUC3* expression in SAMs.

---

## [Author Response]

*In the discussions between the reviewers and editors, a few key points emerged. Please pay particular attention to these:1) A key aspect of your paper is your claim that* STM *expression is mechanosensitive. However, given the considerable lag between the mechanical insult and the change in* STM *expression, the reviewers feel that alternative interpretations are still valid and the claim that the* STM *promoter is mechanosensitive is overstated. We would therefore like to ask you to perform a more comprehensive analysis to strengthen your idea. For example, how does* STM *expression react to transient mechanical stimuli, such as pinching?*

We agree with this comment. In the strictest sense, saying that *STM* promoter is mechanosensitive is an overstatement as it may suggest a direct relation to actors of the mechanotransduction pathways, even though the *STM* promoter takes several hours to be activated. We find that the *STM* promoter is rather the downstream/indirect target of a mechanotransduction pathway. We had highlighted the indirect nature of the induction at the end of our initial submission and we have now stated this earlier in the revised text.

Here we need to dissect the meaning of mechanotransduction: the transduction of mechanical stress is achieved by various pathways, and these pathways highly depend on the nature of the mechanical stimulus. For instance, touch is an extrinsic, discontinuous mechanical perturbation that triggers thigmorphogenesis, and could for instance be coded in frequency. As an example, repeated stem bending can lead to stem thickening and this response can also lead to desensitization (e.g. Martin et al., 2010 J. Exp Bot). Another kind of mechanical perturbation is the intrinsic and continuous membrane tension in animal cells, leading to the inhibition of endocytosis, cytoskeleton reorganization and cell polarity (e.g. Houk et al., 2012 Cell). Tension also builds up in cell walls in plants: the slow extension of cell wall under turgor pressure, leading to wall thinning, triggers wall synthesis to compensate for wall weakening (Cosgrove 2005 Nat. Rev. Mol Cell Biol). Therefore mechanical cues can be either discontinuous or continuous in nature. The relevant mechanotransduction (and timing) will highly depend on the nature of these cues. Our data on *STM* are rather consistent with a response to intrinsic continuous tension than with discontinuous touch, and we have reworded the text in this revision to clarify this point.

The role of intrinsic/continuous mechanical stress in gene expression starts to be uncovered in plants. For instance, the expression of the *ELA1* gene has recently been shown to depend on the presence of mechanical signals in the growing immature seeds. In this context, mechanical perturbations induce a change in expression after 24 hours of prolonged exposure to stress (Creff et al*.,* 2014 Nat. Com.).

We tried pinching meristems. In fact, this can serve as a negative control for our ablation experiments: when a needle touches the surface without breaking the top cell wall, *STM* is not induced. This means that *STM* is induced only when high and prolonged tensile stresses are present. A similar response was observed for microtubule orientation in the meristem: microtubule reorient only when the top wall breaks, i.e. when continuous directional tensile stresses are induced (Hamant et al., 2008 Science). This observation is more in line with *STM* expression pattern maxima, which are not stochastic, are slow to change and correspond to regions under high and continuous tensile stresses. This means that *STM* could be the downstream target of the wall integrity pathway for instance (continuous stress), rather than the touch mechanotransduction pathway (intermittent stress). Consistent with this, we also tested the potential impact of jasmonic acid on *STM* induction and could not detect any significant induction (see below).

Note that we observed that the induction of *STM* expression scaled to the size of the ablation; furthermore, in addition to ablations, we also performed compressions and isoxaben experiments, and these (continuous) mechanical perturbations also induce *STM*.

In this revision, we have added *STM* in situ hybridization after ablation to confirm our finding, we have analyzed other boundary-expressed genes to demonstrate the specificity of the *STM* response to stress, and we have tested the role of JA in *STM* response. All these new data are consistent with our initial claim. See below for more details.

*Moreover, we would like to ask you to perform experiments to exclude the possibility that the observed expression changes are due to a wounding response. Analysis of* STM *expression upon local jasmonate application is one experiment that can be helpful here. Demonstrating that the behavior of the* STM *reporter is unchanged in the JA signaling mutant* coi1 *would be another option. (We are aware that others have suggested that JA may be involved in thigmotropism, but it appears that those effects are probably rather indirect).*

This is a good suggestion. We have applied JA (100 μM) on *pSTM::CFP* meristems and did not see any significant induction (while our *pJAZ10: :GUS* and *pVSP2: :GUS* positive controls were induced by such JA treatments, done in parallel). This confirms that *STM* is not induced by wounding stress, and also is more consistent with an absence of link with thigmomorphogenesis. Note that if compression and isoxaben treatments are likely to induce microcracks in cell walls, they do not induce wounding or cell death, yet they all lead to *STM* induction. We therefore believe that (continuous) mechanical stress is the most likely common factor in these different tests.

*2) There are some questions about the specificity of your observations. If there is indeed causality linking mechanical stress and* STM *expression rather than correlation only, one could expect that this does not hold for any gene expressed in the boundary region. Performing the key experiments with other genes that are expressed in the boundary region could answer this question. Such control experiments should focus on other transcriptions factors.*

In our initial submission, we had presented the (absence of) response of *PINOID* as such a negative control, to show that induction by stress is indeed specific to *STM*. Yet, we agree with the reviewer that a single gene, and not even a transcription factor, is too weak to fully support our claim. In this revision, we present the response to mechanical perturbations for two new boundary-expressed transcription factors. We decided to focus our efforts on *CUC1* and *CUC3*. Both genes display a boundary specific expression, albeit with minor differences both in their exact spatio-temporal pattern and regulation. We found that *CUC1* expression is completely insensitive to ablations, thus confirming that all boundary-expressed transcription factors are not induced by mechanical perturbations. In contrast, we found that *CUC3* is always induced after ablation, and we even found that this induction happened earlier than *STM*. Interestingly, *CUC3* has a greater contribution to the separation of cotyledons than *CUC1* and *CUC2* (Vroemen et al., 2003 Plant Cell) and it displays a slightly different spatio-temporal expression pattern that of *CUC1* and *CUC2. CUC3* is also not regulated by miRNA164. We thus thank the reviewer for this suggestion, as our manuscript is now strengthened with the example of two boundary-expressed genes from the same family, one being induced by mechanical perturbation while the other is not.

*3) Please also carefully address some technical issues. It would be especially important to demonstrate that your assumption that the* amiRNA *construct preferentially downregulates* STM *in the boundary region is true. Combination of this construct with a* STM *protein fusion reporter could substantiate this claim.*This is indeed a good point and we have performed the required experiment. We had introgressed the *pSTM::STM-Venus* marker in the *pSTM::STMamiRNA* lines beforehand and in this revision, we provide the corresponding images: we can indeed detect a down regulation of STM-Venus signal in the *pSTM::STMamiRNA* lines, albeit at various levels, depending on the lines. In these lines, the *pSTM::STM-Venus* signal becomes more homogeneous, consistent with a down‐regulation of *STM* expression at the boundary (as seen on top and side views in Figure 3).

Following the suggestion by the reviewers, we also further characterized the *stm-dgh6 pSTM::STM-Venus* line in much more details. We now show the late phenotype of *stm-dgh6* plants in which vegetative growth can be observed. More importantly, we demonstrate that full complementation occurs in homozygous *stm-dgh6* lines when the *pSTM::STM-Venus* construct is also homozygote. We apologize for the confusion in the initial submission and thank again the reviewers for raising these issues; we believe that these extra data strongly support our initial claim.

Reviewer #1:*[…] In summary, curvature and* STM *reporter expression are correlated in the shoot apical meristem. However, the experiments in this paper do not satisfactorily solve the question of causation. Because* KNOX *genes can cause alterations in curvature in otherwise smooth surfaces (e.g. Tsiantis lab work in Cardamine leaf; Long and Barton, 1998, Development, for work in the* Arabidopsis *embryo), the simplest explanation seems to be that* STM *causes the alterations in curvature by inhibiting cell division and/or cell expansion at the boundary. A mechanism for this is suggested by findings by the Hake lab showing that KNOX targets include genes involved in auxin transport as well as genes involved in cell wall synthesis.*

We agree with the reviewer: this is indeed the simplest explanation and we had already briefly mentioned it in our initial submission. To support it more strongly, we have included the suggested references with the corresponding text and stated that *STM* may very well cause the alteration in curvature, without a mechanical feedback from curvature onto *STM* expression.

Here we rather view the high degree of correlation between *STM* expression and curvature as a necessary condition for the existence of a feedback from shape onto *STM* expression. In other words, we certainly do not deny the simplest scenario, we instead investigate whether a strong correlation can reflect the presence of additional cues, such as auxin depletion or mechanical stress, that would reinforce the role of *STM* in shaping the boundary, by channeling its expression.

*As for the experiments showing that mechanical stress induces* STM *expression, it is possible that many types of wounding or perturbations to the shoot apical meristem will cause an indirect increase in* STM *reporter expression as the plant attempts to compensate for the insult. The authors should test additional stresses to see if this is a more general phenomenon.*

As detailed above, we now clarify the definition of mechanical stress perception (continous vs. discontinuous) and we show that jasmonic acid, a known product of wounding, does not induce *STM* expression in the meristem. We also want to underline that in addition to ablations and compressions, we also observed an induction of *STM* expression after isoxaben treatment, which corresponds to a very different way to increase tension in cell walls.

*The authors should also extend the analysis to other "meristem genes" expressed especially in the boundary zone –* CUCs, BOPs, JLO *– to determine if this is specific to* STM*.*

As discussed above, we now show that *CUC1* expression is not induced after ablations, while *CUC3* expression is. This represents a major addition to the initial submission, as this demonstrates that in the boundary, transcription factors, even from the same family, can be clearly discriminated according to their ability to respond to mechanical stress.

Reviewer #2:

*The core message of this very interesting manuscript is that mechanical stress activates the expression of* STM*, a homeodomain transcription factor with key roles in maintaining meristem fate at the boundary of meristem and organ primordial and that* STM *is required for boundary formation and organ separation. The finding that an important developmental regulator is under control of physical forces in the growing organism while at the same time acting on shaping the organism is certainly of relevance to a wide audience. The experiments (mostly live cell imaging of fluorescent reporters, coupled to perturbations) and analyses shown are of high quality and are mostly well presented. However, in the present form the manuscript has several important shortcomings, which need to be addressed, but based on the exciting findings, I think it is a strong candidate for publication in* eLife *after revision.1) The authors exclusively use a fluorescent reporter as proxy for* STM *activity, however, it does not faithfully recapitulate endogenous* STM *expression. The differences need to be addressed, preferably by in situ hybridization of reporter plants using* STM *and CFP probes. I think this experiment is essential validate the major claims concerning the responsiveness of* STM *expression to mechanical stimuli.*

In this revision, we provide in situ hybridization of *STM* after ablations, and in which we can detect a stronger *STM* signal around the ablation site, thus confirming the results obtained in our two fluorescent *STM* marker lines. Note however, that we cannot obtain dynamic behavior and thus the induction level cannot be quantified; only the correlation between the spatial pattern of induction and the position of the ablation is relevant. Furthermore, we now provide evidence that the *pSTM::STM-Venus* line can fully complement the *stm-dgh6* strong allele, confirming that the higher *STM* promoter activity at the boundary that we see in our *STM* marker lines is biologically relevant. Last, the analysis of other boundary-expressed genes such as *PID* and *CUC1*, which do not respond to mechanical perturbations, also further strengthen the results obtained in *STM*, by demonstrating specificity.

*Along these lines, I would also like to see that endogenous* STM *expression responds to a selected experimental stimulus, such as pinching. Again, these experiments could be done in the reporter line, comparing* STM *and CFP expression by qRT-PCR. These experiments would unequivocally demonstrate that the observed CFP induction is not an artifact of the reporter.*

As discussed in the main text above, pinching does not induce *STM* expression in our hands. We believe that the response of *STM* to ablation, compression or isoxaben treatment rather reflect a response of *STM* to continuous, intrinsic, growth-related stress. *STM* being such an essential gene, it is likely that its expression should be relatively insensitive to stochastic cues and should instead be channeled to ensure reproducibility. In this scenario, pinching would be filtered out in normal conditions, while growth patterns may be used to channel *STM* expression. In this revision, we have clarified the type of mechanical stress we are dealing with in the main text, notably by comparing the *STM* response in the SAM to that of cortical microtubules. In both cases, a steady response (*STM* promoter increased activity, MT hyperalignment) is caused by intrinsic continuous stresses, and not by touch. The absence of *STM* induction after jasmonate exposure further supports this conclusion.

*2) The boundary phenotype in the* pSTM::STM *rescue and* amiRNA *silencing plants is not documented properly. To be able to judge the phenotypes, we need SAM microscopy and curvature measurements. The pictures shown in Figure 3 are not very informative with regards to the boundary effect, rather they show overall growth habitus.*

We fully agree with the reviewer and acknowledge that our initial analysis was too superficial. As detailed above, we have improved the presentation of the results with more pictures and close‐ups, together with genotyping protocols. The *STM* amiRNA lines exhibit small meristems and organ fusions, consistent with a primary role of the boundary in organ separation. In this revision, we also measured curvature in the *STM* amiRNA lines. We found that curvature did not scale to meristem size and that the smaller size of the meristem also brought adjacent boundaries close to one another. These two factors might explain the presence of fusions in the *STM* amiRNA lines.

*3) The boundary specific silencing supposed to be happening in the* amiRNA *lines is not documented. Again, in situ hybridization seems essential to analyze where* STM *silencing occurs, to be able to interpret the results shown in Figure 4.*

Same here: this was indeed a missing piece in our initial submission and we have corrected it by providing images of *pSTM::STM-Venus* expression in the amiRNA lines. *STM* is indeed downregulated at the boundary in these lines.

*4) The claim the* STM *responds to mechanical stimuli after 8 hours onwards is not substantiated by the data presented. From the images shown in Figure 5 it seems that the first effects become visible only after around 20 hours. Please comment.*

This was indeed unclear in our initial submission. The earliest induction for *STM* is 8 hours based on the compression experiments. We have not conducted extensive kinetics analysis to check when on average *STM* is induced after mechanical perturbations, but we never saw an induction before 8 hours so far. This has been clarified in this revision.

*5) The authors routinely refer to "*STM *expression" when they are talking about* pSTM::CFP *reporter activity. Please be more careful with wording.*

We now provide *STM* in situ hybridizations after ablation and we also demonstrate that *pSTM::STM-Venus* fully rescues the *stm* mutant, so it is very likely that our claim is also valid for *STM* expression. Yet, we agree with the reviewer and we have clarified this point in this revision.

*6) The claim that* STM *reporter expression is much higher after removal of NPA is not substantiated by the data shown in Figure 1—figure supplement 1.*

Here we meant that after removal of NPA, *STM* reporter expression switches from being relatively homogeneous in the SAM to being high in a new boundary (as the absence of NPA triggers organogenesis), mimicking the pattern observed in dissected meristems expressing the *pSTM::CFP* construct in absence of NPA.

Reviewer #3:*This interesting manuscript by Hamant and coworkers addresses the question of how the expression of* STM*, a key regulator of the shoot apical meristem also required for organ separation, is regulated. This is an important biological problem, which is explored by the authors using a range of sophisticated imaging, image processing and genetic tools. Overall this manuscript is very well written and the data provide strong experimental support for a role of mechanical stress in controlling* STM *expression patterns.*

*1) To analyze the relationship between mechanical stress, and more specifically Gaussian curvature, and* STM *expression, the authors mapped meristem curvature at high resolution and projected this onto a map of* pSTM::CFP-N7 *expression. These experiments convincingly show that negative Gaussian curvature is correlated with enhanced* STM *expression in the boundary domain of the SAM.*

We thank the reviewer for this positive feedback.

*2) The authors further analyze whether* STM *expression in the boundary is functionally important for organ separation during development. To this end,* STM-Venus *preferentially expressed in the boundary was introduced into a strong* STM *mutant (*dgh6*). The resulting transgenic lines exhibited only a partial recovery of meristem maintenance but a complete loss of organ fusion defects.* pSTM::STMamiRNA2 *was introduced into WT to preferentially silence* STM *expression in the meristem boundary. This resulted in reduced, but functional meristems, and major organ fusions.*

*These are creative attempts to specifically alter* STM *expression in the boundary region; however, no evidence is presented to actually confirm that such preferential alteration has actually occurred. It would have been informative to show the expression pattern of* pSTM::STM-Venus *not only in a WT background (Figure 3) but in the* dgh6 *mutant line, as the expression pattern of the weakly expressing WT parent line (Heisler et al. 2005a) may not necessarily have been maintained after crossing.*

As detailed above, the suggested experiment is now included in this revision and confirms that complementation is associated with *STM‐Venus* expression in homozygous *stm-dgh6* lines.

As shown in the initial submission, we find that the *STM* amiRNA lines exhibit small meristems and organ fusions, but no meristem termination, consistent with a primary role of the boundary in organ separation. In this revision, we measured curvature in the *STM* amiRNA lines. We found that curvature did not scale to meristem size and that the smaller size of the meristem also brought adjacent boundaries close to one another. These two factors might explain the presence of fusions in the *STM* amiRNA lines.

*Similarly, it would be useful to know whether* STM *expression is preferentially silenced in the boundary in* pSTM::STMamiRNA2 *lines. Transformation of the constructs into* pSTM::STM-Venus *lines would have been informative. If these experiments are not done, the limitations should at least be discussed.*

As detailed above, we have introgressed *pSTM::STM‐Venus* in the *pSTM::STMamiRNA2* lines and found a down‐regulation of Venus signal in boundaries, when compared to WT plants.

*3) The authors convincingly demonstrate that enhanced* STM *expression in the boundary is correlated with but not dependent on local auxin depletion. Treatment with 2,4-D globally increased auxin levels in the SAM (as monitored using DII-Venus) but did not alter the pattern of* STM *expression (although slightly increasing the intensity of* pSTM::CFP-N7 *expression).*

*One question this experiment raised for me was whether 24 hour incubation with 2,4-D altered meristem organization in any noticeable manner, i.e. were Gaussian curvatures in the boundaries affected by the treatment? According to the authors' model, this should have affected* STM *expression.*

We could not detect a major impact of the treatment on the overall shape of the meristem. 2,4‐D is likely to impact growth rate as a whole, but it is unclear whether this would also increase differential growth. The independence with auxin is unequivocally shown in Figure 7 which STM expression can be induced in a pin1 background, in which no induction of the DII sensor can be detected.

*4) The authors provide strong evidence that mechanical perturbations (compression of meristem and local cell ablation) and pharmacological treatments resulting in altered mechanical stress (isoxaben, NPA+oryzalin) resulted in increased* STM *expression. Based on the authors' initial experiments,* STM *expression was correlated with negative Gaussian curvature. Does this relationship hold when the SAM is perturbed by the various treatments described here? This analysis could greatly strengthen the concept that* STM *expression is regulated by a specific type of mechanical stress.*

We have not measured the relation between curvature and *STM* expression after mechanical perturbation, notably because we would need to include a membrane marker (like FM4-64) that may interfere with the response. We prefer to keep the mechanical perturbations as simple as possible to avoid multiple controls. Yet, we analyzed the correlation between curvature and *STM* expression in the katanin mutant, which displays a delay in tissue folding, and found that the correlation between tissue folding and *STM* expression was maintained. We believe that this experiment addresses at least partly the suggestion from reviewer 3.

[Editors' note: further revisions were requested prior to acceptance, as described below.]

All reviewers and the editors agree that your work is interesting and original, but as you can see from their comments, the reviewers still have some questions about the causality relations put forward in your study. Yet, there is also consensus that this is a hard chicken-and-egg type of problem, and a completely unambiguous answer is hard to come by with current means. Although things might turn out somewhat different in the long run, your study presents one of the best ways forward into this area for the moment, and is likely to stimulate further research. In light of this, we believe that it would be in your own interest to be very careful how you phrase your conclusions. Please go once more through your manuscript and tone down any strong statements with respect to situations that entail a possibly circular logic. In that respect, please pay particular attention to the comments of reviewer 1.

Thank you for this feedback – we have rephrased our conclusions accordingly in this revision.

*Please also clarify the following: there are some doubts whether the experiments show jasmonate insensitivity of* STM *expression as stated in the text. In the supplemental figure, there is clear* STM *expression increase that does not occur in any of the mock treated samples. It thus appears simply wrong to say it is insensitive, and these results should not be "hidden" in the supporting information.*

In the main text, we stated that “we did not observe any significant induction of *pSTM::CFP-N7* signal in most of the plants (N = 10/14, Figure 5–figure supplement 3), and slight fluctuations in CFP signal in the remaining ones (N= 4/14, Figure 5–figure supplement 3). As *pSTM ::CFP* is always induced, we thus thought that wound-induced jasmonate is not the most likely candidate as a secondary messenger between stress and *STM* expression. Our initial statement “wound-induced jasmonate production does not seem to be involved in the induction of *STM* expression after ablation” is indeed an overstatement and we have rephrased it as follows:

“Based on these results, we cannot rule out completely that jasmonate interferes with *STM* expression. Yet, as this contrasts with the systematic and steady induction of *STM* after ablation and with the robust induction of a *pJAZ10::GUS* reporter by jasmonate (Figure 5), wound-induced jasmonate is not the most likely candidate as a secondary messenger between stress and *STM* induction.”

We also have moved the jasmonate results in the main figure.

Reviewer #1:

*This paper presents several interesting observations. My problem with the paper – even after the changes and in some cases even more so after the authors' additions – is that the authors shoehorn their observations into a model in which mechanical stress increases* STM *expression.In the body of the paper, the authors include alternative explanations for the correlation such as in this paragraph:"The fact that* STM *expression correlates with curvature at the boundary can simply be explained by the fact that* STM *reduces the growth rate at the boundary, leading to tissue folding.* KNOX *genes can cause alterations in curvature in otherwise smooth surface (Long and Barton, 1998, Barkoulas et al., 2008). The impact of* STM *on crease formation at the boundary could be mediated by inhibiting cell growth in this domain. Consistent with this scenario, KNOX target genes include genes involved in auxin transport as well as genes involved in cell wall synthesis (Bolduc et al., 2012). However the strong correlation between* STM *promoter activity and tissue folding suggests that a signal, which is related to curvature, could add robustness to the* STM *expression pattern at the boundary."Why would a strong correlation lead one to prefer one direction of causality over the other?*

In the quoted text, we state that hypothesis 1: “*STM* slows down growth and this induces folding” is actually a fact. We are therefore not preferring one direction or the other, we say that direction 1 “*STM* -> shape” exists for sure. We are exploring whether the other direction “shape-> *STM*” also exists. In other words, what we are looking for is whether the correlation between folding and *STM* expression also reflects the presence of a feedback. This does not mean that a feedback exists, we only say that the existence of a correlation is a necessary condition for the feedback to exist. Therefore we need to quantify curvature and relate it to *STM* expression to check whether the correlation is strong or not, before moving on to auxin and mechanics as potential feedback factors. If we had found a very weak correlation between *STM* expression and curvature, then *STM* would not have been on our candidate list for a target of shape feedback.

*We know that* STM/KNOX *expression exists in the absence of tissue folding and also precedes tissue folding in the embryo, and precedes tissue folding in the development of leaf lobing. To my knowledge, there is no case of tissue folding that precedes* STM/KNOX *expression.*

We observed and measured that *STM* promoter activity increases with tissue folding, but it is very likely that *STM* induction indeed precedes tissue folding (although this would be more difficult to quantify, as this would involve measuring regional differential growth – see below). In fact, when comparing *STM* expression at the boundary with the microtubule response to mechanical stress in the same domain, we also see that microtubule align in the boundary before folding occurs, and this alignment can be correlated with differential growth (and thus directional stress) at the boundary before folding: the fast growing initium is generating stress before it forms a crease at the boundary (see e.g. Burian et al., 2013 J. Exp Bot). Here we only focus on curvature because the formation of a crease reflects the presence of stress at the boundary for sure, while a flat shape may or may not be related with stress (differential growth then also needs to be measured). The induction of *STM/KNOX* in leaves and embryo before folding may very well be related to differential growth (and thus stress) preceding folding. We have added this point in the new Discussion.

*Of all the treatments that the authors subject the plants to, pinching would seem to best simulate folding but pinching the meristem does not increase* STM *expression. This argues strongly for the arrow of causation to be in the direction counter to what the authors propose. (The authors make arguments about how pinching is somehow different than the stresses that would normally be sensed by* STM *but I am not convinced).*

We don’t think that the pinching experiment reflects what happens at the boundary: in this domain, mechanical stress is predicted to slowly increase and to become more and more directional as the tissue folds. This is quite different from a transient compression from the top of the tissue. In fact, the ablations, lateral compressions and isoxaben treatments that are included in our manuscript are more consistent with the situation at the boundary: a prolonged exposure to mechanical stress. This discussion comes down to signal transduction kinetics: if one could increase very transiently auxin concentration (for a fraction of a second), the TIR1 auxin pathway would probably not be induced significantly enough to have a visible, steady, impact on gene expression. But this may be sufficient to induce visible short term responses, like a calcium peak. In other words, the cell responds differentially to transient and prolonged cues. Similarly, cellulose synthesis occurs in response to prolonged exposure to mechanical stress (caused by progressive wall thinning/weakening). To our knowledge, one pinching event does not induce a permanent increase of cellulose synthesis; only the repetition of such deformations (like the wind) may have a significant impact on growth. Last, in the text we refer to other documented genes that are downstream targets of mechanical stress. While some of them (TOUCH genes) are clearly induced within minutes of mechanical deformation, others (e.g. ELA1 in the immature seed (Creff et al., 2014 Nat. Com)) take several hours to be induced after prolonged exposure to stress, like STM in our experiments.

*Whatever the reality may be, in this paper alternative interpretations are downplayed. There is no mention of alternative models in the Discussion and the title is overly strong in its assertion given the weak, and by the authors' own admission, indirect, connection between physical stress and* STM *expression.*To clarify again: the fact that *STM* slows down growth and induce folding is not an alternative hypothesis, it is a fact. Here, based on the strong correlation between folding and *STM* expression, we explore whether signals may in turn enhance *STM* expression at the boundary. We explore two signals: auxin and mechanical stress. Other signals could be checked (the list is very long and beyond the scope of this article); this is where alternative hypotheses could be found. We find that if auxin depletion and mechanical stress both correlate with *STM* induction at the boundary, they are not coupled: mechanical induction of *STM* is most likely not mediated via auxin depletion or jasmonate induction. We have added two paragraphs in the Discussion to explore other scenarios (please see: “Alternative cues may be involved in the promotion of *STM* expression […] these may be better candidates for the regulation of *STM* expression at the boundary”) and carefully rephrased our conclusions and title to avoid any misunderstanding.

*Unfortunately, without the framework of mechanical stress acting to increase* STM*, the experiments presented in this paper lack cohesion.*

What is missing for sure is a mutant that would be mechano-insensitive and in which *STM* expression pattern at the boundary would be less robust. We are working on it currently and we hope to provide more conclusive results in the future. Yet, at this point, we believe that this work is a solid analysis of the relation between shape and *STM* expression in the meristem, from which we find that mechanical stress contributes to *STM* expression, in parallel to auxin depletion. We also provide a number of open prospects in the Results and Discussion that should stimulate further research.

Comments on revisions:

*1) The authors shore up their complementation data. They have now used PCR to genotype homozygous* stm/stm*;* pSTM::STM-Venus/pSTM::STM-Venus *double mutants segregating from among the progeny of* stm/+pSTM::STM-Venus/pSTM::STM-Venus *parents. It would have been more convincing if they had recovered a fertile double homozygote but if the PCR data are compelling, then they likely have identified complemented individuals. (Unfortunately, the PCR data aren't shown. I would be worried about using the presence/absence of a 2.6 kb band for determining the presence of the wild type allele as lack of a band can be due to simple PCR failure. Presumably the authors have adequately controlled for this.)*

We have collected seeds from double homozygote thus confirming full complementation.

*2) The authors have added a test for response of* STM *expression to jasmonate, a hormone associated with wounding stress. This is an experiment to determine whether the response of* STM *expression to ablation might be in response to wounding. The data are given in the supplemental section (Figure 5–figure supplement 3). Most plants do not respond – however a significant fraction, 4 out of the 20 or so plants tested,* did *show an increase in* STM *expression in response to jasmonate. The authors dismiss this positive result as unimportant but I am not sure what the basis for their dismissal is.(I can easily envision an entirely different historical scenario for these experiments that would lead to a different set of conclusions: A set of jasmonate researchers could have reasonably concluded that applied jasmonate can increase* STM *expression. They would then have followed up that observation with ablation experiments on the meristem finding that* STM *expression increased in response to wounding.)In short, the jasmonate experiments indicate that the situation is likely to be complex. Rather than sweeping this complexity under the rug, the authors should discuss it.*

Here we are not investigating the contribution of jasmonate in *STM* expression (otherwise indeed 4 inductions out of 20 would have been an interesting lead). We only want to know whether, in response to ablation, the reproducible induction of *STM* is only due to jasmonate production. As jasmonate is systematically induced following ablations and as we see induction of *STM* expression in 4 out of 20 treated meristems and no clear/steady induction in the remaining 16, this is not the most probable hypothesis and indeed a more complex scenario must be called upon. We have rephrased our conclusion to better reflect this and we now show the relevant data in Figure 5. Our initial statement “wound-induced jasmonate production does not seem to be involved in the induction of *STM* expression after ablation” is indeed an overstatement and we have rephrased it as follows:

“Based on these results, we cannot rule out completely that jasmonate interferes with *STM* expression. Yet, as this contrasts with the systematic and steady induction of *STM* after ablation and with the robust induction of a *pJAZ10::GUS* reporter by jasmonate (Figure 5), wound-induced jasmonate is not the most likely candidate as a secondary messenger between stress and *STM* induction.”

We also have moved the jasmonate results in the main figure.

*3) The authors add data for other boundary-expressed genes and find that one –* CUC3*, but not the other,* CUC1*, respond to ablation with increased expression. This is very interesting but I am not entirely sure what to make of it. Again, it reinforces the conclusion that the situation is complex.*

The situation is complex and in this case there is an interesting perspective to follow: *CUC1* expression is regulated by miR164 whereas *CUC3* is not. This suggests that different mechanisms ensure the robustness in expression for members of the same gene family. For this article, the goal was rather to show that only a fraction of boundary-expressed genes are regulated by mechanical perturbations. This is the only conclusion we draw from these extra results.

*(I would add here as well that these experiments are done in NPA treated meristems and therefore the results hold for a context in which auxin is improperly localized. Since there is reason to believe that auxin and* KNOX *genes interact at some level, this is important to keep in mind. A different result might be obtained in the context of properly localized auxin).*

In these experiments, meristems are recovering from NPA treatments so this would be unlikely, but not impossible. Yet, since controls and assays are conducted in the exact same conditions throughout, the conclusions still stand. Note that *STM* induction was also observed in the complete absence of NPA in a *pin1* mutant (Figure 7) and in the continuous presence of NPA (Figure 7—figure supplement 3), so at least *STM* induction by mechanical perturbations does not seem to be NPA dependent.

Reviewer #2:

I am very happy with the thorough revision of the manuscript and the addition of many meaningful and well-executed new experiments. I now support publication of the manuscript without hesitation.

*The only issue I still have is the rather poor quality of* STM *in situ hybridizations after ablation. With some good will it is possible to identify enhanced* STM *RNA levels next to the ablation site, but it could be helpful to include more apices (if the authors have pictures available) to make that trend clearer.*

We agree with the reviewer that the enhanced signal is rather subtle. We have added another picture of in situ hybridization as requested. We also have added a word of caution: “Note that ablation may also provide increased accessibility to the probe.”

We also have increased the number of replicates for the ablations on the *CUC1* marker line (all negative for induction as previously shown).